# Climate signals in stable carbon and hydrogen isotopes of lignin methoxy groups from southern German beech trees

**Anna Wieland**[1], **Markus Greule**[1], **Philipp Roemer**[2], **Jan Esper**[2,3], **and Frank Keppler**[1,4]

[1]Institute of Earth Sciences, Heidelberg University, Im Neuenheimer Feld 234–236, 69120 Heidelberg, Germany
[2]Department of Geography, Johannes Gutenberg University Mainz, 55128 Mainz, Germany
[3]Global Change Research Institute of the Czech Academy of Sciences (CzechGlobe), 60300 Brno, Czech Republic
[4]Heidelberg Center for the Environment (HCE), Heidelberg University, 69120 Heidelberg, Germany

**Correspondence:** Frank Keppler (frank.keppler@geow.uni-heidelberg.de) and Anna Wieland (anna.wieland@geow.uni-heidelberg.de)

**Abstract.** Stable hydrogen and carbon isotope ratios of wood lignin methoxy groups ($\delta^{13}C_{LM}$ and $\delta^2H_{LM}$ values) have been shown to be reliable proxies of past temperature variations. Previous studies showed that $\delta^2H_{LM}$ values even work in temperate environments where classical tree-ring width and maximum latewood density measurements are less successful for climate reconstructions. Here, we analyse the annually resolved $\delta^{13}C_{LM}$ values from 1916–2015 of four beech trees (*Fagus sylvatica*) from a temperate site near Hohenpeißenberg in southern Germany and compare these data with regional- to continental-scale climate observations. Initial $\delta^{13}C_{LM}$ values were corrected for the Suess effect (a decrease of $\delta^{13}C$ in atmospheric $CO_2$) and physiological tree responses to increasing atmospheric $CO_2$ concentrations considering a range of published discrimination factors. The calibration of $\delta^{13}C_{LM}$ chronologies against instrumental data reveals the highest correlations with regional summer ($r = 0.68$) and mean annual temperatures ($r = 0.66$), as well as previous-year September to current-year August temperatures ($r = 0.61$), all calculated from 1916–2015 and reaching $p < 0.001$. Additional calibration trials using detrended $\delta^{13}C_{LM}$ values and climate data (to constrain effects of autocorrelation on significance levels) returned $r_{summer} = 0.46$ ($p < 0.001$), $r_{annual} = 0.25$ ($p < 0.05$) and $r_{prev.September-August} = 0.18$ ($p > 0.05$). The new $\delta^{13}C_{LM}$ chronologies were finally compared with the previously produced $\delta^2H_{LM}$ values of the same trees in order to evaluate the additional gain of assessing past climate variability using a dual-isotope approach. Compared to

$\delta^{13}C_{LM}$, $\delta^2H_{LM}$ values correlate substantially more strongly with large-scale temperatures averaged over western Europe ($r_{prev.September-August} = 0.69$), whereas only weak and mainly insignificant correlations are obtained between precipitation and both isotope chronologies ($\delta^{13}C_{LM}$ and $\delta^2H_{LM}$ values). Our results indicate the great potential of using $\delta^{13}C_{LM}$ values from temperate environments as a proxy for local temperatures and, in combination with $\delta^2H_{LM}$ values, to assess regional- to sub-continental scale temperature patterns.

## 1 Introduction

Trees are a powerful archive in global climate research (Esper et al., 2016, 2018; Ljungqvist et al., 2020), as they endure for centuries in widespread temperate climate zones and form yearly growth rings that can be used to analyse factors influencing wood formation (Stoffel and Bollschweiler, 2008). Weather and climate parameters affect, either directly or indirectly, the physiological process within these tree rings. Thus, growth-specific parameters – such as ring width, maximum latewood density, and the isotopic composition of bulk wood, cellulose, or lignin – have proven to be climate sensitive (McCarroll and Loader, 2004; Daux et al., 2011; Esper et al., 2015; Esper, 2000; Reynolds-Henne et al., 2007; Hafner et al., 2011; Konter et al., 2014; Kress et al., 2010; Treydte et al., 2001; Wang et al., 2011). In addition Greule et al. (2019) and Keppler et al. (2007) showed that stable hydrogen and carbon isotope ratios of wood lignin

methoxy groups ($\delta^2$H$_{LM}$ and $\delta^{13}$C$_{LM}$ values) have great applicative potential for paleoclimate reconstructions. For a more detailed overview of its applications in paleoclimate research we refer the reader to the studies of Anhäuser et al. (2020), Gori et al. (2013), Greule et al. (2021), Hepp et al. (2017), Lee et al. (2019), Lu et al. (2020), Van Raden et al. (2013), and Wang et al. (2020). The isotopic ratios of these specific chemical moieties (-OCH$_3$ groups) of lignin remain unchanged throughout the lifetime of a tree and thus reflect the methoxy isotopic composition at the time of its biochemical formation (Greule et al., 2008; Keppler et al., 2007). While the traditional methods for analysing stable isotope ratios of alpha-cellulose or nitrate cellulose require time-consuming preparation, $\delta^{13}$C$_{LM}$ and $\delta^2$H$_{LM}$ values can be readily measured as iodomethane (CH$_3$I) upon treatment with hydroiodic acid, providing a fast and straightforward preparation process (Greule et al., 2008). Furthermore, the removal of water from bulk wood samples is not necessary, as it does not affect the isotope analysis (Greule et al., 2008, 2009). It is also advantageous for isotope analysis that methoxy groups are highly abundant in tree rings, as wood contains around 25 %–30 % of lignin and the proportion of methoxy groups in lignin (on a carbon basis) may reach 20 % (Keppler et al., 2007). Therefore, small sample amounts of only 1 and 2.5 mg of bulk wood (milled or in pieces) are required for reliable measurements of $\delta^{13}$C$_{LM}$ and $\delta^2$H$_{LM}$ values (Greule et al., 2008, 2009). Finally, the analytical procedure for measuring $\delta^2$H$_{LM}$ values was considerably improved by the availability of new reference materials that are in full accordance with the requirements of normalising stable isotope measurements (Greule et al., 2020, 2019).

The studies by Gori et al. (2013) and Mischel et al. (2015) compared the stable carbon, hydrogen, and oxygen isotopic composition of whole wood, cellulose, and lignin methoxy groups. The study by Gori et al. (2013) used tree samples from three different elevation sites in the south-eastern Alps (900, 1300, and 1900 m), while the study of Mischel et al. (2015) used trees from a low elevation environment (about 300 m altitude). Both studies showed that the stable carbon and oxygen (Mischel et al., 2015) or stable carbon, oxygen, and hydrogen (Gori et al., 2013) isotope values of whole wood and cellulose are strongly correlated, while the carbon and hydrogen isotope composition of lignin methoxy groups correlate to a lesser extent with the other components. On this basis, both studies suggest that the stable carbon and hydrogen isotope values of methoxy groups contain a different climate signal and seem to be influenced by different environmental and biochemical factors. The extraction of cellulose, however, may not be necessary, as the isotopic compounds of cellulose and whole wood receive the same climate signal. The measurement of lignin methoxy groups, on the other hand, could be a new and additional proxy for climate reconstructions in a different temporal and spatial context.

This conclusion can be supported by the study of McCarroll et al. (2003), which implicated that the key to amplifying the climate signal lies in combining independent proxies that are not similar. In this context, Gori et al. (2013) showed that the best prediction model for reconstructing temperature changes is obtained when the hydrogen and carbon isotope compounds of whole wood and methoxy groups are combined. For a detailed discussion regarding the stable isotopic compounds of cellulose, whole wood, and lignin methoxy groups, we refer the readers to the studies by Gori et al. (2013) and Mischel et al. (2015). Most previous methoxy-based research CE1 has applied $\delta^2$H$_{LM}$ values for climate studies (Anhäuser et al., 2017a, b; Riechelmann et al., 2017; Anhäuser et al., 2020; Greule et al., 2021; Keppler et al., 2007; Wang et al., 2020). In general, the hydrogen isotopic composition of trees is controlled by its source water and hence the stable isotopes composition of the local precipitation (Tang et al., 2000; Sternberg, 2009). Therefore, the temperature-dominated signal in $\delta^2$H$_{precipitation}$ (Dansgaard, 1964) is reflected in $\delta^2$H$_{LM}$ values as has been demonstrated for mid-latitude sites (Greule et al., 2021; Anhäuser et al., 2017b). Recently, Greule et al. (2021) discussed the known biosynthetic pathways involved in the formation of lignin methoxy groups and applied a simple numeric model to explain the observed biosynthetic isotope fractionation pattern between $\delta^2$H$_{precipitation}$ and $\delta^2$H$_{LM}$ values. For a detailed description, we refer the reader to the study by Greule et al. (2021).

A highly significant correlation was documented between $\delta^2$H$_{LM}$ values, mean annual $\delta^2$H$_{precipitation}$ values ($r = 0.66$), and mean annual temperatures (MAT), whereas "shifted" annual $\delta^2$H$_{precipitation}$ values and MAT (defined as previous September to recent August) showed the highest coefficients with $r = 0.73$ and $r = 0.56$ (Anhäuser et al., 2020). Wang et al. (2020) found significant correlations between $\delta^2$H$_{LM}$ values and April–August temperatures ($r = 0.58$–0.7) for two coniferous species (*Larix gmelinii*, larch & *Pinus sylvestris* var. *mongolica*, pine) from a permafrost forest in northeastern China. Even higher correlations were reported between beech trees from a low elevation site in southern Germany and western European large-scale temperature changes at $r = 0.72$ (Anhäuser et al., 2020). Therefore, it is assumed that stable water isotopes in precipitation mainly indicate large-scale atmospheric phenomena rather than changes in local or regional climate states (Anhäuser et al., 2020; Bowen et al., 2019).

Although there exist some studies (Riechelmann et al., 2016; Mischel et al., 2015; Gori et al., 2013; Wang et al., 2020), less attention has been given to the climate sensitivity of $\delta^{13}$C$_{LM}$ values. The carbon of each annual tree ring has its origin in the atmospheric CO$_2$. Thus the carbon isotope composition in trees mainly consists of the isotopic values of atmospheric CO$_2$ ($\delta^{13}$C$_{atmos}$), the concentration of atmospheric CO$_2$, the diffusion and fractionation of $\delta^{13}$C through stomatal pores ($-4.4$ mUr), and carbon fixation via

the photosynthetic enzyme Rubisco ($-27$ mUr) (Francey and Farquhar, 1982). Please note that we follow the suggestion by Brand and Coplen (2012) and express isotope $\delta$-values in milli-Urey (mUr), after Urey (1948), instead of per mil (‰). The carbon isotopic composition in trees can be expressed as the deviation between $\delta^{13}C_{atmos}$ and the isotopic discrimination of $^{13}C$ during carbon diffusion and fixation by plants (Eq. 1) (Keeling et al., 2017):

$$\delta^{13}C_{tree} = \delta^{13}C_{atmos}$$
$$- \left[ a + (b-a)\frac{c_i}{c_a} - \frac{(b-a_m)\left(\frac{A}{c_a}\right)}{g_i} - f\Gamma^*/c_a \right], \quad (1)$$

where $a$ expresses the fractionation by diffusion through the stomata and $b$ describes the fractionation to carboxylation, while $c_i$ reflects the inner leaf $CO_2$ concentration and $c_a$ the ambient air $CO_2$ concentrations (Francey and Farquhar, 1982). The last two terms of Eq. (1) represent the mesophyll and photorespiration effects, with $a_m$ representing the fractionation by dissolution and diffusion from the intercellular air spaces to the sites of carboxylation in the chloroplasts, $A$ the leaf-level gross photosynthesis, $g_i$ the mesophyll conductance, $f$ the discrimination due to photorespiration, and $\Gamma$ the $CO_2$ compensation point in the absence of day respiration (Cernusak et al., 2013; Seibt et al., 2008; Farquhar et al., 1982; Keeling et al., 2017). There are large uncertainties about these variables, and the effects are normally neglected; however, they are necessary for understanding the discrimination changes of $^{13}C$ due to increasing $CO_2$ concentrations (Seibt et al., 2008). The terms of mesophyll and photorespiration are both negative, and their absolute magnitudes decrease with increasing $c_a$, resulting in increasing discrimination with rising $CO_2$ concentrations (Keeling et al., 2017).

Since the fractionations due to stomatal conductance and Rubisco ($a$ and $b$) are considered constant and since the terms of mesophyll and photorespiration are normally negligible, the discrimination of $^{13}C$ is mainly controlled by the $c_i/c_a$ ratio. If $c_i$ increases, stomatal control limits the rate of photosynthesis. The dominance of stomatal control and photosynthesis rate thereby depend on various environmental factors, including temperature, air humidity, precipitation amount, and seasonality (McCarroll and Loader, 2004). The carbon isotopic signatures of plant materials can be further modified by post-photosynthetic fractionations. Differences in $\delta^{13}C$ values between plant organs arise if fractionation appears during transport of metabolism transport or if respiratory fractionation changes in different organs (Fung et al., 1997; Badeck et al., 2005; Seibt et al., 2008). As a result, leaves are usually enriched in $^{12}C$ compared to other plant organs (Yoneyama et al., 1997). Since the isotopic fractionation between the photosynthate and cellulose or lignin is assumed to be small (Francey and Farquhar, 1982) and since post-photosynthetic processes are not currently

understood (Cernusak et al., 2009), the $\delta^{13}C_{LM}$ values of this study were not further corrected for the leaf-to-wood offset; consequently, the environmental factors that influence the $c_i/c_a$ ratio are also considered to additionally control $\delta^{13}C_{LM}$ values.

The few studies that applied the $\delta^{13}C_{LM}$ values of tree rings already demonstrated a relationship with climate parameters. Wang et al. (2020) and Riechelmann et al. (2016) documented highly significant correlations between $\delta^{13}C_{LM}$ values and mean summer temperatures in high elevation environments. Wang et al. (2020) observed the highest correlations with April to August temperatures ($r = 0.64$) and Riechelmann et al. (2016) with June to August temperatures ($r = 0.66$). In addition, Gori et al. (2013) report correlations with spring and annual mean temperatures, and Mischel et al. (2015) with August maximum temperatures. However, in all previous studies, non-significant correlations ($p > 0.05$) were reported with precipitation.

In this study, we evaluate the applicability of the $\delta^{13}C_{LM}$ values of trees as a paleoclimate proxy in temperate, low elevation environments. Therefore, we measured the annually resolved $\delta^{13}C_{LM}$ values of four *Fagus sylvatica* L. trees in southern Germany at Hohenpeißenberg and analysed the climate sensitivity and non-climatic response (to atmospheric $CO_2$ changes) of the $\delta^{13}C_{LM}$ values. Furthermore, to evaluate the potential of reconstructing past climate variability using a dual-isotope approach, we revisit the $\delta^2H_{LM}$ values of the same trees provided by Anhäuser et al. (2020). However, these previous data were corrected according to new constraints regarding analytical issues of the isotope measurements of methoxy groups (Greule et al., 2021). Finally, the dual isotope methoxy measurements of Hohenpeißenberg tree rings were used to critically evaluate their potential as a proxy for regional- to sub-continental scale temperature patterns.

## 2 Material and methods

### 2.1 Study site

The study site is located close to the Hohenpeißenberg municipality in southern Germany, where tree samples were collected from the northeastern slope of the Hoher Peißenberg mountain ($47°48'$ N, $11°01'$ E; altitude $\sim 800$ m). For a detailed map of the sampling site, we would like to refer the reader to the study by Anhäuser et al. (2020). The region is characterised by a strong temperature increase, particularly since the 1980s, and insignificant precipitation trends (Fig. 1a). Annual mean temperatures range from $6.22°C$ (1940) to $9.73°C$ (2018), and precipitation totals from $788$ mm (1943) to $1316$ mm (1966). The seasonal climate is characterised by a distinct precipitation peak in summer, including $138$ mm (period 1961–1990) in July (Fig. 1b).

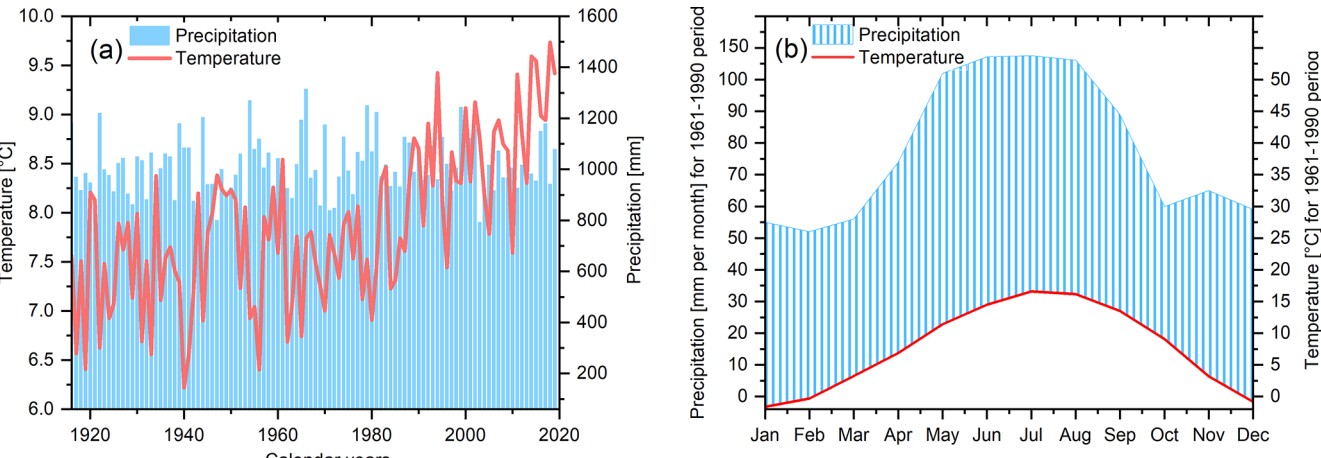

**Figure 1.** Climate at Hohenpeißenberg. (**a**) Mean annual temperatures and monthly precipitation totals since 1916, and (**b**) mean monthly temperatures and precipitation amounts calculated from 1961–1990 (CRU TS 4.04 data at 47.75° N/11.25° E).

## 2.2 Tree samples and $\delta^{13}C_{LM}$ analysis

Four *Fagus sylvatica* trees (F1–F4) were sampled in spring and autumn 2016. The samples were extracted at breast height (1.2 m above ground) using an increment borer with a diameter of 5 mm. Every tree is represented by two cores, with the age of each tree dating back to 1912 (F1), 1858 (F2), 1890 (F3), and 1916 (F4). Since each tree covers the period 1916–2015, further examinations were focused on this time interval.

For the determination of $\delta^{13}C_{LM}$ values, the modified Zeisel method was used (Keppler et al., 2007, 2004; Greule et al., 2009). The method is based on the reaction between methyl ethers or esters and hydriodic acid (HI) to form iodomethane (Zeisel, 1885). In a 1.5 mL crimp glass vial, 250 µL HI (57 wt % aqueous solution, Acros; Thermo Fisher Scientific) were added to the 1–10 mg annually dissected tree rings. The vials were sealed with crimp caps and heated for 30 min at 130 °C. Samples were then equilibrated at room temperature (22 °C) for at least 30 min before an aliquot of headspace was injected into the gas chromatograph combustion isotope ratio mass spectrometry (GC-C-IRMS) analytical system. An amount of 10–90 µL of headspace was injected via an autosampler (A200S, CTC Analytics, Zwingen, Switzerland) with a split injection of 10 : 1 to the HP 6890 N gas chromatograph (Agilent, Santa Clara, USA). The gas chromatograph was fitted with a DB-5MS, Agilent J&W capillary column (length 30 m, internal diameter 0.25 mm, film thickness 0.5 µm) with an initial oven temperature of 50 °C for 2.9 min and a ramp at 50 °C min$^{-1}$ to 110 °C. Helium was used as a carrier gas at a constant flow rate of 1.8 mL min$^{-1}$. Using an oxidation reactor (ceramic tube ($Al_2O_3$), length 320 mm, internal diameter 0.5 mm) with Cu, Ni, and Pt wires inside (activated by oxygen) and a reaction temperature of 960 °C, $CH_3I$ is oxidised to $CO_2$. Before the $CO_2$ flows through a GC

Combustion III Interface (ThermoQuest Finnigan) into the isotope ratio mass spectrometer (253 Plus 10 kV IRMS, Thermo Fisher Scientific), the accrued water was removed through a semipermeable membrane (NAFION®). A tank of high purity carbon dioxide (carbon dioxide 4.5, Messer Griesheim, Frankfurt, Germany) was used as the monitoring gas. For all values, the delta ($\delta$) notation is used, employing the term Urey (Ur, after Urey, 1948) as the isotope delta value unit (Brand and Coplen, 2012). Hence, 1 mUr equates to 1 ‰.

The $\delta^{13}C_{LM}$ values were normalised by a two-point linear calibration using two reference materials, namely potassium methyl sulfate (HUBG 2) and beech wood (HUBG 4), as described by Greule et al. (2019, 2020). The $\delta^{13}C_{LM}$ values of HUBG 2 and HUBG 4 were calibrated against international isotope reference material (V-PDB) with an isotopic value of $\delta^{13}C_{VPDB} = +1.60 \pm 0.12$ mUr for HUBG2 (Greule et al., 2019) and $\delta^{13}C_{VPDB} = -30.17 \pm 0.13$ mUr for HUBG4 (Greule et al., 2020). Before and after every sixth measurement, HUBG 2 and HUBG 4 were measured alternately. The tree rings of the two cores of F1 were measured as triplicates ($n = 3$). Differences between the triplicates were always less than 1 mUr with an average deviation of 0.08 mUr. The maximum differences between two individual cores of the same tree ranged from 1.54 mUr for F1 to 3.26 mUr for F2. Since each tree is represented by the average of two cores, the variances between the triplicate measurements of F1 are marginal compared to the much larger differences of the two cores of the same tree. Therefore, to drastically reduce analytical costs, further tree rings from F2–F4 were analysed by single measurements.

## 2.3 Correction of $\delta^{13}C_{LM}$ values for non-climatic environmental factors

Due to anthropogenic burning of fossil fuels, the atmospheric $CO_2$ concentration is steadily increasing. Since fossil $CO_2$ has a lighter carbon isotopic composition than the atmosphere, the $\delta^{13}C$ values in atmospheric $CO_2$ ($\delta^{13}C_{atmos}$) show a prominent downwards trend. This so-called "Suess effect" (Keeling, 1979) describes a decrease in the $\delta^{13}C_{atmos}$ value from $-6.41$ mUr in 1850 to a current value of $-8.6$ mUr in 2020. Consequently, leaf-internal $CO_2$ ($c_i$) is already depleted in $^{13}C$, and even more $^{12}C$ can be assimilated in leaf sugars, yielding to more negative $\delta^{13}C_{LM}$ values. As this decline is a non-climate effect, the carbon isotopic composition of tree rings needs to be corrected by adding the differences for each year between the $\delta^{13}C_{atmos}$ and the pre-industrial value ($-6.41$ mUr) to the measured $\delta^{13}C_{LM}$ values (Suess effect corrected values are declined as $\delta^{13}C_{LM\_S}$) (McCarroll and Loader, 2004). Here, the $\delta^{13}C_{atmos}$ series was obtained from McCarroll and Loader (2004) and the Mauna Loa Observatory (Keeling et al., 2005a; https://scrippsco2.ucsd.edu/data/atmospheric_co2/mlo.html, last access: 4 February 2022). Furthermore, the leaf-internal $^{13}C$ discrimination increases with rising $CO_2$ concentrations while the absolute magnitude of mesophyll and photorespiration decreases (Keeling et al., 2017). It is important to note that there is no pre-defined way to correct the $\delta^{13}C$ values of trees due to increasing $CO_2$ concentrations. However, in our study, the Suess effect corrected $\delta^{13}C_{LM\_S}$ values were multiplied considering a correction factor per increasing $CO_2$ parts per million by volume (ppmv) compared to the pre-industrial $CO_2$ concentrations. We used the $CO_2$ series from MacFarling Meure et al. (2006) and the Global Monitoring Laboratory NOAA (https://gml.noaa.gov/ccgg/trends/, last access: 1 February 2022; https://scrippsco2.ucsd.edu/, last access: 1 February 2022). Different studies proposed diverse correction factors. For instance, Kürschner (1996) suggested a correction value of $0.0073$ mUr ppmv $CO_2^{-1}$, Treydte et al. (2009) of $0.012$ mUr ppmv $CO_2^{-1}$, Wang et al. (2011) of $0.016$ mUr ppmv $CO_2^{-1}$, Feng and Epstein (1995) of $0.02$ mUr ppmv $CO_2^{-1}$, and Riechelmann et al. (2016) of $0.032$ mUr ppmv $CO_2^{-1}$. The physiological response due to increasing $CO_2$ concentrations may be different between tree species and locations and is itself a current and important field of study.

We additionally detrended the $\delta^{13}C_{LM}$ data using 30-year cubic smoothing splines to emphasise high-frequency variations and to evaluate the effects of autocorrelation in our analyses. The resulting $\delta^{13}C_{LM\_high\text{-}frequency}$ data were compared with (30-year spline) detrended temperature data to ensure that significant correlations are not simply related to the warming trend prevalent over the last 60 years (Fig. 1a).

## 2.4 Correction of $\delta^2H_{LM}$ values considering new reference material

The $\delta^2H_{LM}$ values of trees from Hohenpeißenberg presented by Anhäuser et al. (2020) were normalised using two $CH_3I$ reference standards. As $CH_3I$ is a different material compared to wood and since the two $CH_3I$ reference standards did not cover the entire range of the $\delta^2H_{LM}$ values of the samples ($-295$ to $-224$ mUr), this study applied the newly available reference material investigated and recommended by Greule et al. (2019, 2020). Thus, previously measured $\delta^2H_{LM}$ values were corrected using the suggested equation of Greule et al. (2021) (Eq. 2). Accordingly, the corrected data shifts the previous $\delta^2H_{LM}$ series to more positive values, and the differences between previous and corrected $\delta^2H_{LM}$ series become larger with decreasing $\delta^2H_{LM}$ values.

$$\delta^2H_{LM\_corrected}\,[mUr] = \left(\delta^2H_{LM}\,[mUr] \times 0.78\right) \\ - 45.71\,mUr \qquad (2)$$

Please note that the previous $\delta^2H_{LM}$ chronology of Anhäuser et al. (2020) included nine cores. F1 was represented by three cores and F2–F4 by two cores. To have the same number of replicates of all four trees ($n = 2$), one core from tree F1 was removed.

## 2.5 Climate data and statistics

The sensitivity of $\delta^{13}C_{LM}$ and $\delta^2H_{LM}$ chronologies to climate were assessed by comparing the mean $\delta^{13}C_{LM}$ and $\delta^2H_{LM}$ anomalies as deviations from the 1961–1990 mean with monthly resolved temperatures and precipitation data from a nearby grid point at $47.75°$ N and $11.25°$ E as well as large-scale gridded temperatures using the latest CRU TS version 4.04 via the KNMI climate explorer (Trouet and Jan van Oldenborgh, 2013; Harris et al., 2020). For correlations with detrended $\delta^{13}C_{LM\_low\text{-}frequency}$ [TS1] values, low-frequency variances from the temperature data were removed using a 30-year cubic smoothing spline. Correlations among the $\delta^{13}C_{LM}$ and $\delta^2H_{LM}$ tree-ring series and between climate parameters and isotope chronologies were calculated over the period of 1916 to 2015 using the Pearson correlation coefficient ($r$). Temporal changes between proxies and climate parameters were assessed using 31-year running correlations. Here, $p$ values $< 0.05$ were considered significant, and $p < 0.001$ highly significant. The reconstructive capabilities of $\delta^{13}C_{LM}$ and $\delta^{13}C_{LM\_high\text{-}frequency}$ chronologies were assessed using Durbin–Watson statistics (DW), testing lag-1 autocorrelation in the linear model residuals, the reduction of error (RE), and coefficient of efficiency (CE) statistics. Any positive values of RE and CE are indicative of adequate capabilities for the reconstructions (Cook et al., 1994; Briffa et al., 1988). All data analyses, statistics, and graphs were calculated and plotted using the software Arstan, OriginPro 2021, and R.

## 3   Results

### 3.1   $\delta^{13}C_{LM}$ values, correction for the Suess effect, and physiological response

The $\delta^{13}C_{LM}$ values of the four tree series range from $-32.66$ to $-26.02$ mUr from 1916 to 2015 (Fig. 2a). The $\delta^{13}C_{LM}$ anomalies (deviations from the 1961–1990 mean) range from $-3.18$ to $3.34$ mUr with a standard deviation $\sigma = -0.05 \pm 1.05$ mUr (Fig. 2b). The data are characterised by relatively low inter-series correlation (Rbar) of $r = 0.23$ ($p > 0.05$) and include a systematic change in coherency over time as indicated by 31-year moving Rbar values. Rbar reaches an average $r$ of 0.24 at the beginning of the chronology between 1916 and 1939 followed by a rapid decrease to minimum values of $r = 0.02$ in 1948 and then a gradual increase to maximum $r = 0.23$ in 1995 (Fig. 2c). The expressed population signal (EPS) equals 0.71 and is thus lower than the commonly accepted threshold of $\geq 0.85$. However, among others, the study of Buras (2017) critically evaluated the threshold of 0.85 for the EPS value and concluded that the application of the EPS value does not necessarily provide valid information on whether tree-ring data can be used for temperature reconstructions.

The mean $\delta^{13}C_{LM}$ chronology comprises four trees with two cores each over the past 100 years (Fig. 3 black solid line) and is characterised by two phases. From 1916 to 1965, the chronology includes a minor positive trend of 0.006 mUr yr$^{-1}$, followed by a negative linear trend of $-0.02$ mUr yr$^{-1}$ from 1966 to 2015 (Fig. 3 dashed lines). After the correction of the Suess effect, the mean $\delta^{13}C_{LM\_S}$ values are less negative and shifted by $+0.3$ mUr in 1916 and $+2.04$ mUr in 2015 (Fig. 3 yellow line). Applying additional corrections that account for physiological response due to increasing atmospheric $CO_2$ concentrations, the $\delta^{13}C_{LM}$ values further increase to more positive values. This effect is particularly visible from the second half of the 20th century until today and clearly depends on the value that has been used as the correction factor (Fig. 3 pink, red, and green line). This study used a wide range of the previously applied correction factors of Treydte et al. (2009) ($\delta^{13}C_{LM\_T}$), Feng and Epstein (1995) ($\delta^{13}C_{LM\_FE}$), and Riechelmann et al. (2016) ($\delta^{13}C_{LM\_RL}$). After the correction of the Suess effect, Rbar values initially decreased from 0.25 for uncorrected $\delta^{13}C_{LM}$ values to 0.16. The highest inter-series correlations were recorded for the $\delta^{13}C_{LM\_RL}$ series ($r = 0.55$, $p < 0.001$), where the increase of the correlation factor is simply related to the addition of a simulated trend to the series. The effect of autocorrelation on Rbar values is mitigated in the 30-year high pass filtered data ($\delta^{13}C_{LM\_high\text{-}frequency}$). Mean Rbar reduces to 0.22 and the lag-1 autocorrelation decreases from 0.652 for the uncorrected $\delta^{13}C_{LM}$ to 0.046 for $\delta^{13}C_{LM\_high\text{-}frequency}$ series. Moving Rbar values of the $\delta^{13}C_{LM\_high\text{-}frequency}$ indices show a similar trend to the inter-series correlations of the raw $\delta^{13}C_{LM}$ series, characterised by a strong depression between 1940 and 1966 (Fig. 2c dashed line).

### 3.2   Climate sensitivities of $\delta^{13}C_{LM}$ chronologies

When comparing $\delta^{13}C_{LM}$ chronologies with climate data, we find positive correlations with regional temperatures and largely non-significant correlations with precipitation. The coefficients tend to increase when considering $\delta^{13}C_{LM}$ chronologies that were corrected for the Suess effect ($\delta^{13}C_{LM\_S}$) and physiological response due to increasing atmospheric $CO_2$ concentrations ($\delta^{13}C_{LM\_T} < \delta^{13}C_{LM\_FE} < \delta^{13}C_{LM\_RL}$) (Figs. 4 and S1 in the Supplement). The highest correlation coefficients were found between $\delta^{13}C_{LM}$ values and summer (JJA) temperatures ($\delta^{13}C_{LM\_S}$: $r = 0.52$ to $\delta^{13}C_{LM\_RL}$: $r = 0.68$) ($p < 0.001$; the degrees of freedom were reduced, due to lag-1 autocorrelation), followed by MAT ranging from $r = 0.42$ to 0.66, and "shifted" annual temperatures (previous September to August) ranging from $r = 0.34$ to 0.61. Among the mainly non-significant correlations with precipitation totals, the highest coefficient was identified with the Suess effect corrected $\delta^{13}C_{LM\_S}$ chronology and summer precipitation ($r = -0.33$).

Additional assessments of climate signals using 30-year high-pass filtered chronologies revealed that correlations with summer temperatures decrease to $r = 0.46$ but are still highly significant at $p < 0.001$. The correlation coefficient with MAT (January–December) is lower with $r = 0.25$ ($p < 0.05$) and non-significant with shifted MAT (September$_p$–August) ($r = 0.18$, $p > 0.05$).

Since the study by Anhäuser et al. (2020) found a strong correlation between $\delta^2H_{LM}$ series and large-scale atmospheric phenomena, this study further evaluates the relationships with large-scale seasonal temperatures using $\delta^{13}C_{LM\_RL}$ values (https://climexp.knmi.nl/start.cgi, last access: 23 June 2022). The patterns extend from southern Europe to middle Scandinavia and cover the UK and western Poland, Slovakia, and Hungary. Here, the highest correlations with $r > 0.6$ were found between the $\delta^{13}C_{LM\_RL}$ values and summer temperatures in southern Germany, Austria, northern Italy, and northeastern Spain, followed by somewhat lower correlations of $r > 0.5$ with summer temperatures in middle Germany, Switzerland, and France and with fall temperatures in northeastern Spain (Figs. 5a, S2). Correlations with winter and spring temperatures are always $r < 0.5$ (Fig. S2). The highest correlation between the $\delta^{13}C_{LM\_high\text{-}frequency}$ chronology and large-scale summer temperatures extend from southeastern Germany to middle France and cover Switzerland and northern Italy with $r > 0.4$ (Fig. 5b).

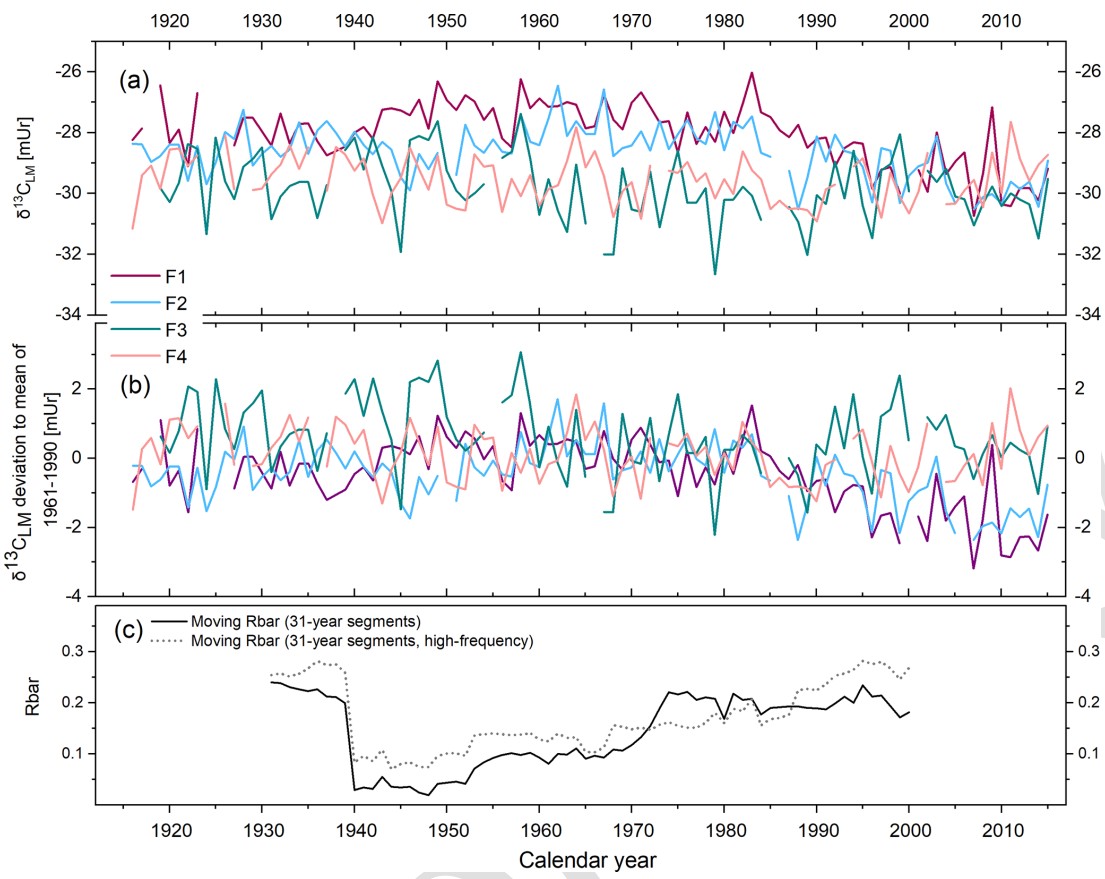

**Figure 2.** $\delta^{13}C_{LM}$ series of four *Fagus sylvatica* trees (F1–F4) from 1916–2015. **(a)** Annually resolved $\delta^{13}C_{LM}$ values. **(b)** $\delta^{13}C_{LM}$ anomalies shown as deviations from the 1961–1990 mean. **(c)** Moving Rbar of the $\delta^{13}C_{LM\_high\text{-}frequency}$ (solid line) and the $\delta^{13}C_{LM}$ (dotted line) series.

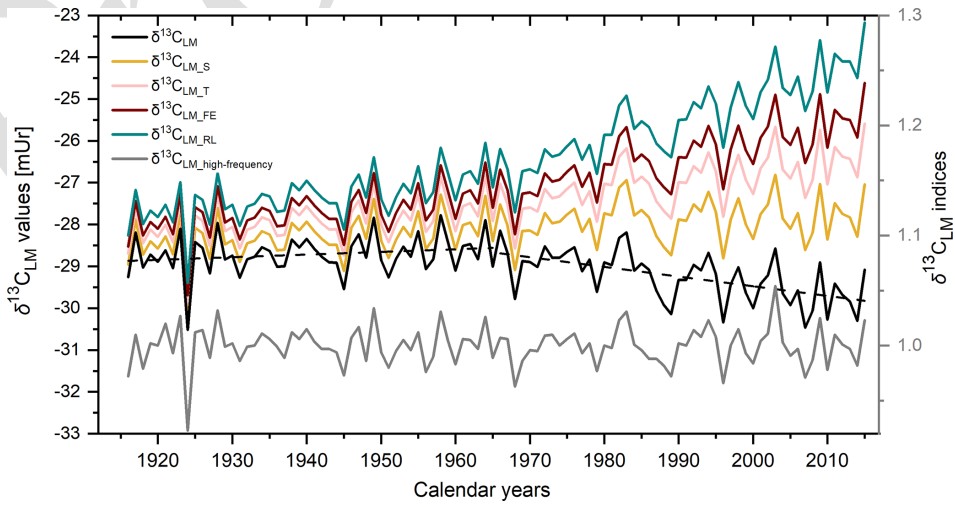

**Figure 3.** The mean $\delta^{13}C_{LM}$ chronology (black line). Dashed lines show the linear trends until 1965 and from 1965 to 2015. $\delta^{13}C_{LM\_S}$ is the chronology after the corrections of the Suess effect, $\delta^{13}C_{LM\_T}$ after additional correction of physiological response due to increasing $CO_2$ concentration with $\delta^{13}C_{LM\_T}$ considering $0.012\,\mathrm{mUr\,ppmv^{-1}}\,CO_2$, $\delta^{13}C_{LM\_FE}$ considering $0.02\,\mathrm{mUr\,ppmv^{-1}}\,CO_2$, and $\delta^{13}C_{LM\_RL}$ considering $0.032\,\mathrm{mUr\,ppmv^{-1}}\,CO_2$. The grey curve shows the detrended $\delta^{13}C_{LM\_high\text{-}frequency}$ indices.

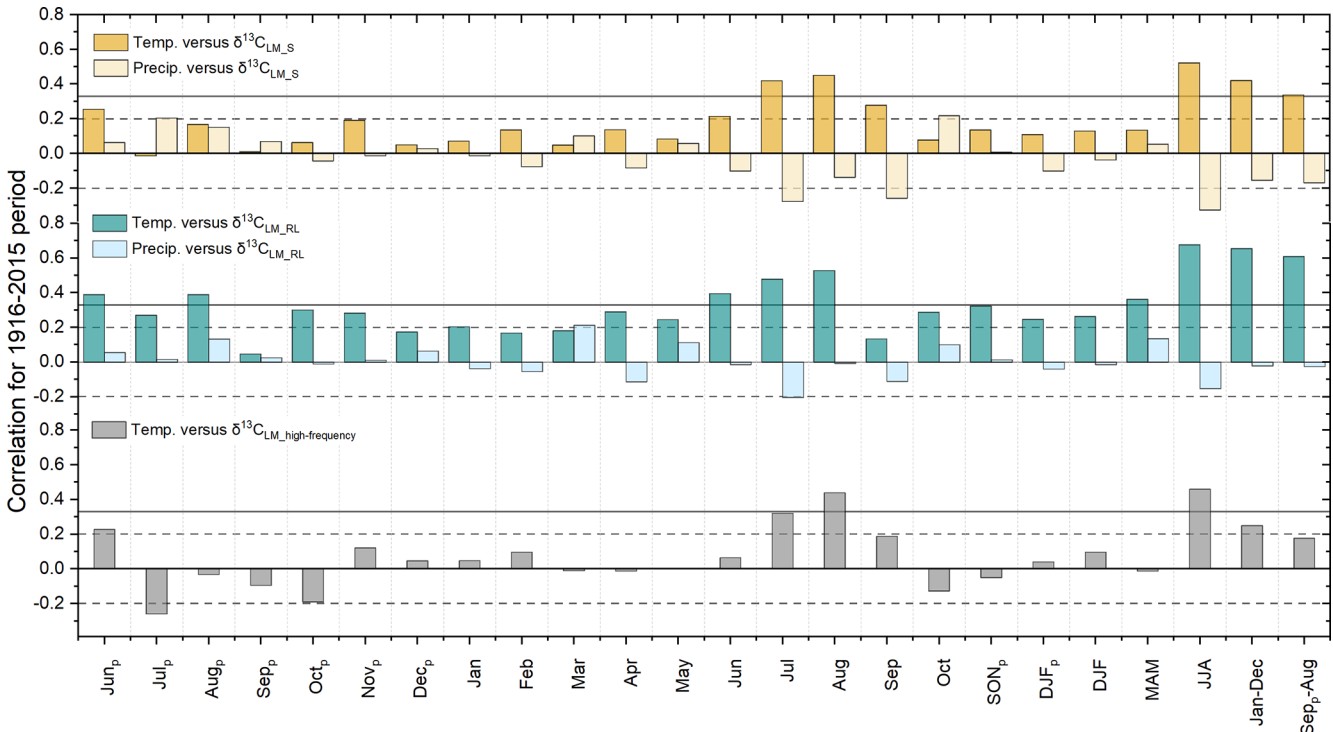

**Figure 4.** Correlation coefficients between corrected $\delta^{13}C_{LM\_S}$, $\delta^{13}C_{LM\_RL}$, $\delta^{13}C_{LM\_high\text{-}frequency}$ chronologies and local temperatures and precipitation totals from 1916 to 2015. The subscript p indicates the months of the previous year, and horizontal lines indicate the significance levels, with solid lines representing highly significant ($p < 0.001$) and dashed lines representing significant values ($p < 0.05$).

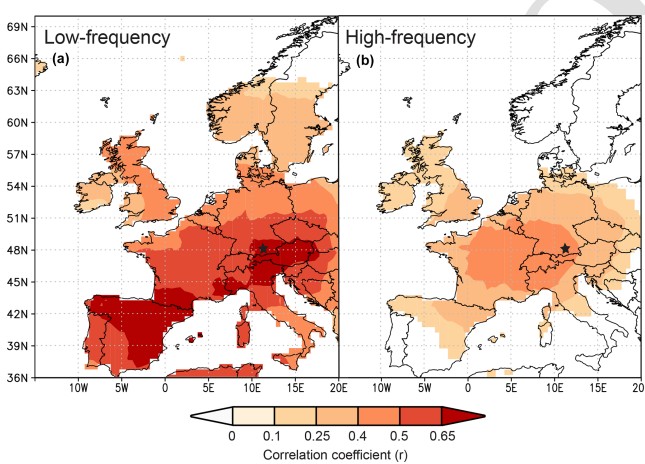

**Figure 5.** Spatial correlations between summer temperatures (CRU TS4.04) and $\delta^{13}C_{LM\_RL}$ anomalies **(a)** and $\delta^{13}C_{LM\_high\text{-}frequency}$ indices **(b)** from 1916–2015. Black star marks the Hohenpeißenberg in Germany.

## 3.3   Running correlations and transfer function

The local summer temperatures were modelled by $\delta^{13}C_{LM\_RL}$ values using a linear regression model following Eq. (3):

$$\left[{}^{\circ}C\right] = \frac{\delta^{13}C\left[mUr\right] + 0.36}{0.83\left[mUr\,{}^{\circ}C^{-1}\right]} \tag{3}$$

$$\left[{}^{\circ}C\right] = \frac{\delta^{13}C_{high\text{-}frequency}\left[mUr\right] - 1.29}{-0.29\left[mUr\,{}^{\circ}C^{-1}\right]}. \tag{4}$$

The regression model residuals range from $-2.13$ to $1.91$ and show an increasing trend of $0.022 \pm 0.002$ over the past 100 years. Between 1916 and 1963, residuals are mainly negative, whereas since 1964 residuals are mainly positive (Fig. 6a, b). Furthermore, we calculated the Durbin–Watson (DW) statistic of the regression model residuals and reveal a weak DW value of 0.86. which indicates a strong positive autocorrelation ($p < 0.001$). If two series are autocorrelated, the effective sample size and thus the degrees of freedom may be reduced. Since the significance of the correlation coefficient depends on the degrees of freedom, significant correlations may well be non-significant under the consideration of autocorrelation (Wigley et al., 1987).

Running correlations between gridded instrumental and modelled regional summer temperatures reveal substantial temporal changes. The 31-year moving correlation values range from 0.03 to 0.09 with an average $r = 0.03 \pm 0.02$ between 1939 to 1965. Before 1939, correlation coefficients

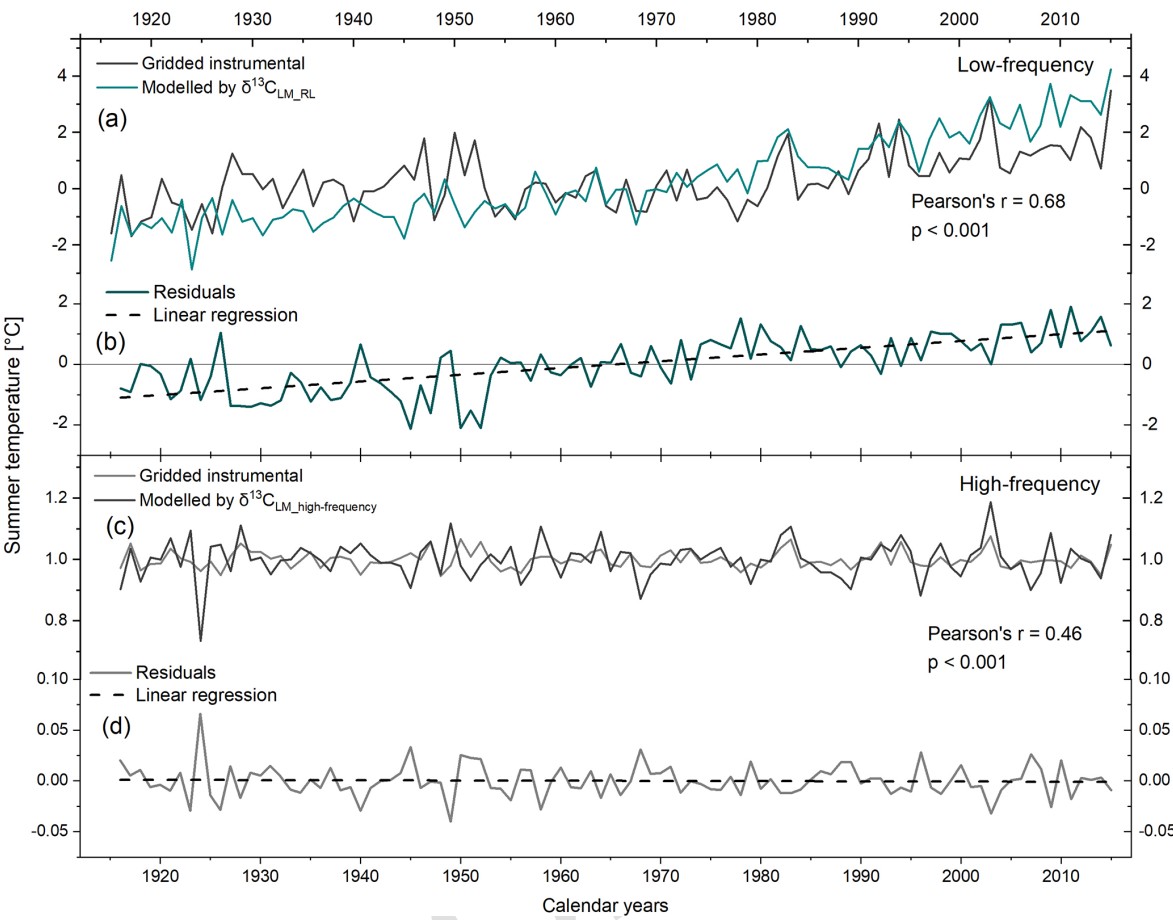

**Figure 6.** Gridded instrumental and modelled JJA temperatures by **(a)** $\delta^{13}C_{LM\_RL}$ values and **(c)** $\delta^{13}C_{LM\_high-frequency}$ indices from 1916–2015. **(b, d)** Plots of residual trends through time for low- and high-frequency.

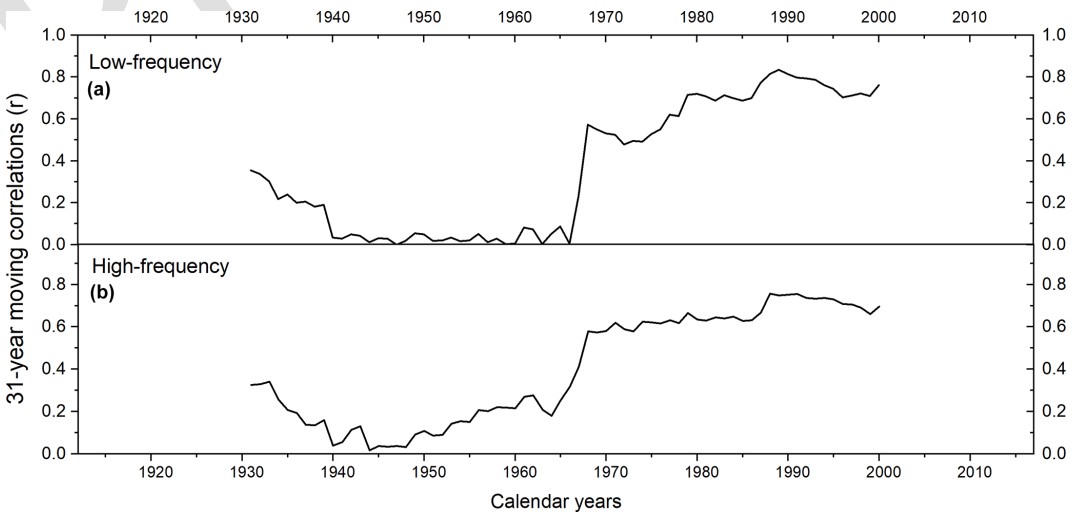

**Figure 7.** 31-year moving correlations between **(a)** low- and **(b)** high-frequency gridded instrumental and modelled summer temperatures.

range from 0.18 to 0.36 with an average $r = 0.25 \pm 0.07$. After 1965, correlation values increase rapidly with $r$ ranging from 0.23 to 0.84 (average $r = 0.66 \pm 0.13$) (Fig. 7a).

To eliminate the effect of autocorrelation and to constrain high-frequency variations, we employed $\delta^{13}C_{LM\_high-frequency}$ indices for modelling summer temperatures using Eq. (4). The high-frequency chronology reveals a DW value of the regression residuals close to the optimum value of 2 (DW = 2.3) suggesting the robustness of the high-frequency signal. The residuals range between $-0.04$ and 0.07 and show no trend over the 1916–2015 period (Fig. 6c, d).

To determine the strength of the relationship between modelled and observed temperatures, we calculated the reduction of error (RE) and coefficient of efficiency (CE) statistics after Cook et al. (1994).

When conducting a split calibration/verification on the high-frequency linear model, a rather weak temporal robustness is indicated (calibration period 1970–2015: RE = $-1.67$ and CE = $-1.68$; calibration period 1916–1968: RE = $-4.73$ and CE = $-4.75$). To explore this issue in more detail, we calculated 31-year moving correlations between gridded instrumental and modelled summer temperature indices. Running correlations showed significant values at the beginning of the chronology, followed by decreasing correlations to $r = 0.04$ in 1944 and then a gradual increase toward $r = 0.76$ in 1990 (Fig. 7b). Hence, summer temperatures are mainly significantly represented in $\delta^{13}C_{LM\_high-frequency}$ indices. However, between 1935 to 1954, moving correlations are non-significant. Recalculating the RE and CE statistics for only the 1956–2015 period, RE and CE values substantially increase and are close to zero (calibration period: 1986–2015: RE = $-0.19$ and CE = $-0.19$; calibration period 1956–1985: RE = $-0.54$ and CE = $-0.54$).

## 3.4  $\delta^2H_{LM}$ chronology and climate signal

The $\delta^2H_{LM}$ values of the tree cores from Hohenpeißenberg previously presented by Anhäuser et al. (2020) were corrected using the revised relationship provided by Greule et al. (2021) (see Sect. 2). The revised data showed maximum and minimum $\delta^2H_{LM}$ values ranging from $-274$ to $-221$ mUr around a mean value of $-246 \pm 9$ mUr ($1\sigma$ standard deviation) and maximum and minimum $\delta^2H_{LM}$ anomalies from $-12.3$ to 19.4 mUr with a mean value of $1.9 \pm 6.4$ mUr. Considering the chronologies of the four trees, a highly significant inter-series correlation of $r = 0.33$ ($p < 0.001$) can be reported, whereby somewhat higher Rbar values are observed between single radii of the same tree ranging from 0.57 to 0.8. Figure 8 (solid black line) shows the mean $\delta^2H_{LM}$ chronology of the four trees over the past 100 years. A linear increasing trend of 0.14 mUr yr$^{-1}$ is observed over the whole period from 1916 to 2015, but it is

also obvious that the trend is more positive (0.38 mUr yr$^{-1}$) from 1970.

The highest $r$ values between the corrected $\delta^2H_{LM}$ anomalies and temperature are recorded for the "shifted" annual temperatures (September$_p$–August) with $r = 0.58$ ($p < 0.001$) and the MAT (January–December) with $r = 0.57$ ($p < 0.001$). Furthermore, summer temperatures and $\delta^2H_{LM}$ values also showed a highly significant correlation of $r = 0.51$ ($p < 0.001$), whereas correlations with winter and previous fall decrease ($r = 0.31$ and $r = 0.3$). Correlation coefficients with precipitation were non-significant ($p > 0.05$). Similar to the results of Anhäuser et al. (2020), the highest correlations were documented with large-scale western European temperatures ($r = 0.69$) averaged for the region of 15° W–20° E and 25–75° N (Fig. 9).

## 4  Discussion

### 4.1  $\delta^{13}C_{LM}$ values, response to atmospheric $CO_2$ changes, and other environmental factors

The average inter-series correlations vary between the uncorrected ($r = 0.23$), the Suess effect corrected ($r = 0.16$), and the maximum physiological response corrected ($r = 0.55$) $\delta^{13}C_{LM}$ values, as they are influenced by different trend changes (Riechelmann et al., 2016). The correction procedure of the Suess effect removes the prevailing long-term decrease from $\delta^{13}C_{LM}$ values, and the correction for the physiological response produces a positive long-term trend. Furthermore, our results showed that the incremental application of corrections for non-climate-related trends due to changes in atmospheric $CO_2$ not only increases the inter-series correlations among trees but also improves the correlations with climate parameters. However, by simply multiplying a correction factor onto the $\delta^{13}C_{LM}$ values, strong (simulated) increasing trends were added to the $\delta^{13}C_{LM}$ series and then related to the temperature, which has, as is generally known, a strong modern trend due to climate change. Therefore, the highest correlation coefficients between the corrected $\delta^{13}C_{LM\_RL}$ series and temperature have to be interpreted with care when considering how plant isotopic discrimination responds to increasing $CO_2$ concentrations. However, as far as there is no greater detail available on how $CO_2$ affects plant isotopic fractionation, we get at least an idea regarding the magnitude in which the physiological responses of beech trees in low elevation environments might be affected by increasing $CO_2$ concentrations. Since the modern increasing trend in the corrected $\delta^{13}C_{LM}$ records is simulated, the series cannot be used to position recent temperature increases within a long-term context.

Correlation coefficients with temperature increase significantly by adding a correction factor that accounts for a strong $CO_2$ response (0.032 mUr ppmv$^{-1}$ as introduced by Riechelmann et al., 2016) (correlations with $\delta^{13}C_{LM\_T} <$

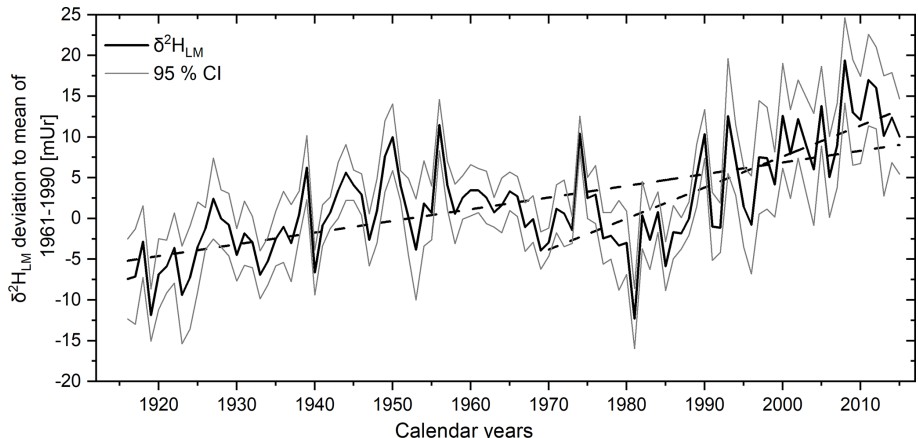

**Figure 8.** The corrected mean $\delta^2 H_{LM}$ anomalies as deviations from the 1961–1990 mean from 1916–2015. The 100-year record (black line) represents the mean of the eight individual tree-ring series with 95 % CI (grey lines). Linear regression lines (dashed) are shown for the whole period and from 1970–2015.

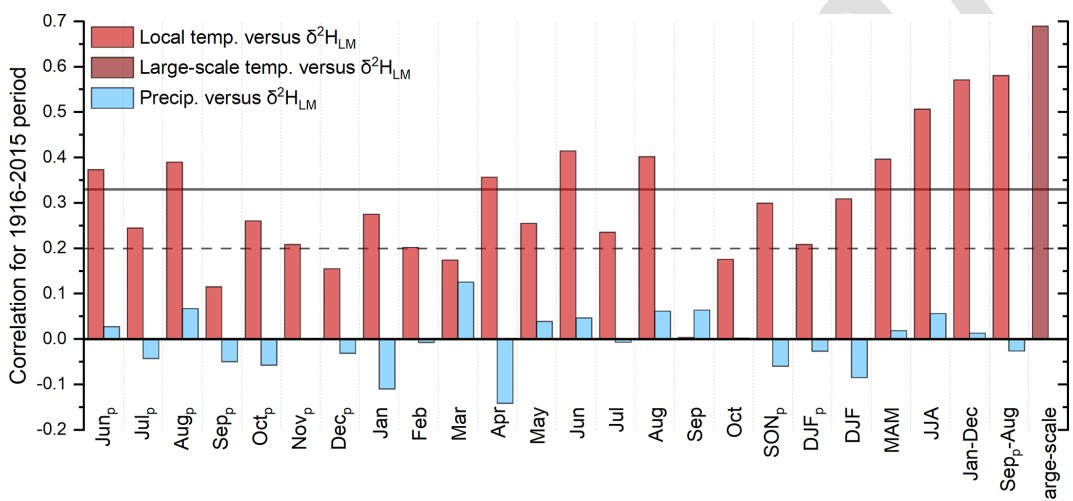

**Figure 9.** Correlation coefficients between the $\delta^2 H_{LM}$ chronology, local and large-scale temperatures, and local precipitation amounts from 1916–2015. The subscript p indicates the months of the previous year, and the horizontal lines indicate the significance levels, with solid lines representing highly significant ($p < 0.001$) and dashed lines representing significant values ($p < 0.05$).

$\delta^{13} C_{LM\_FE} < \delta^{13} C_{LM\_RL}$). The use of this correction factor is amongst the strongest in literature and would constitute the strong $CO_2$ response of trees growing in low elevation environments. This higher mean discrimination of $^{13}$C by *Fagus Sylvatica* seems to be enhanced compared to other tree species, which might be related to the lower water-use efficiency of deciduous trees compared to evergreen conifer species (Riechelmann et al., 2016). The studies by Feng and Epstein (1995) and Treydte et al. (2009) mainly used evergreen conifer species and therefore required lower correction factors.

Further, we would like to point out that temporal changes of $\delta^{13}$C values can not only be explained by climate influences or changes in the isotope ratio and concentration of atmospheric $CO_2$. A number of environmental factors –

such as nitrogen deposition, tropospheric ozone pollution, $SO_x$ deposition, and drought stress as well as water, light, and nutrient availability – might also affect plant isotopic discrimination (Wittig et al., 2009; Guerrieri et al., 2011; Büntgen et al., 2021). For example, the study by Leonardi et al. (2012) indicated a significant correlation between nitrogen deposition and $\delta^{13}$C values in conifers and in angiosperms, while Büntgen et al. (2021) found reduced stomatal conductance as protection from drought stress, which may lead to an increase in $\delta^{13}$C values.

Furthermore, tree-age-related influences on $\delta^{13}$C values (seen in the studies of Brienen et al., 2017; Francey and Farquhar, 1982; Gagen et al., 2008; Vadeboncoeur et al., 2020) have resulted in depleted and linearly increasing $\delta^{13}$C values in the first 50 years of tree growth (Gagen et

al., 2006) due to the uptake of soil-respired $CO_2$ by trees growing close to the forest floor (van der Merwe and Medina, 1991) and an increase in irradiance from the understory to the canopy (Farquhar et al., 1982). To quantify the impact of an age-related trend on our $\delta^{13}C_{LM}$ chronology, a site-specific linear regression analysis between trees of various ages over a common period was calculated. The four trees used in this study cover, in the first 40 years of the chronology (1916–1956), the juvenile (F1, cambial age 12–52), mature (F2, cambial age 81–121), and intermediate (F3, cambial age 34–74, and F4, cambial age 27–67) growing phases. As the youngest tree, F1 shows a positive slope of $0.02\,\text{mUr yr}^{-1}$. For F2, F3, and F4, a minor trend of $-0.001$, $0.001$, and $-0.01\,\text{mUr yr}^{-1}$ was calculated (Fig. S3). The $\delta^{13}C_{LM}$ values of the juvenile growing phase of F3 and F4 were also partly measured (1890–1929 and 1916–1939) and show a negative slope of $-0.013$ and $-0.004\,\text{mUr yr}^{-1}$. Since only F1 shows an increasing $\delta^{13}C_{LM}$ trend in the juvenile stage and since none of the trees showed a depletion in $\delta^{13}C_{LM}$ values, the $\delta^{13}C_{LM}$ chronology seems to be unaffected by an age-related trend. The absence of the juvenile effect could be explained by the low forest density at this study site. Vadeboncoeur et al. (2020) reported strong tree size effects on $\delta^{13}C_{LM}$ values in trees growing in a closed canopy environments, where openly grown saplings were often unaffected and had similar values to large, codominant trees. However, to completely exclude an age-related trend on $\delta^{13}C_{LM}$ values, several trees from study sites with different forest densities should be analysed in further studies.

## 4.2 Climate sensitivity of $\delta^{13}C_{LM}$ values

The greatest climate response was documented between $\delta^{13}C_{LM}$ values and regional summer temperatures ($\delta^{13}C_{LM\_S}$: $r = 0.52$ to $\delta^{13}C_{LM\_RL}$: $r = 0.68$) (Fig. 4). Since correlations with seasonal, large-scale temperatures (western European surface temperatures) also showed the highest correlations with summer temperatures in the surrounding area of the study site (Fig. 5), $\delta^{13}C_{LM}$ values seem to reflect local temperature variations better than large-scale fluctuations (Fig. 11).

The strong temperature response in $\delta^{13}C_{LM}$ values could be at least part of different indirect signals, as several other environmental factors like sunshine, irradiation, relative air humidity (RH), or soil moisture status also strongly correlate with temperature. Relative air humidity is linked to the vapour pressure deficit; both RH and soil moisture status directly control stomatal conductance, whereas irradiance exerts a strong influence on photosynthetic rate (McCarroll and Loader, 2004). However, RH and soil moisture status are also strongly correlated to antecedent precipitation (McCarroll and Loader, 2004). Strong temperature- and weak precipitation response in the $\delta^{13}C_{LM}$ values of this study indicate that $\delta^{13}C_{LM}$ ratios are predominantly controlled by the photosynthetic rate (McCarroll and Loader,

2004). This conclusion can be supported by simultaneously increasing $\delta^{13}C_{LM}$ values and tree-ring width (Fig. S4). For example, a decreased assimilation rate reduces tree growth (Masle and Farquhar, 1988) and increases the $c_i/c_a$ ratio, thus reducing $\delta^{13}C_{LM}$ values (Francey and Farquhar, 1982).

Similar findings were reported in previous studies applying stable carbon isotopes of tree rings (McCarroll and Pawellek, 2001; Treydte et al., 2009; Riechelmann et al., 2016; Gagen et al., 2006). However, most of these studies analysed trees from cold, moist, high latitude, or high elevation sites. Here, we now demonstrate that local temperatures also strongly influence the $\delta^{13}C_{LM}$ ratios of trees growing in mid-latitude, low elevation environments, which are less extreme conditions.

Summer temperature seems to predominantly control $\delta^{13}C_{LM}$ values at the Hohenpeißenberg site, especially in the last 50 years, since moving correlation coefficients between gridded instrumental and modelled records substantially increase after 1966 (Fig. 7). This change in climate sensitivity is not entirely controlled by the increasing temperature trend recorded over recent decades, as similar correlation changes are recorded when using 30-year high-pass filtered tree-ring series ($\delta^{13}C_{LM\_high\text{-}frequency}$) and gridded instrumental data. Similar inferences were reported in the study by Treydte et al. (2009). Here, climate correlations were calculated using high-frequency $\delta^{13}C_{LM}$ series to avoid biases from potentially non-climatic, long-term trends.

The weak DW statistic and positive trend in the low-frequency regression model residuals is partly influenced by the value that has been set as the correction factor on $\delta^{13}C_{LM}$ values. A change in $^{13}C$ discrimination in early increasing $CO_2$ concentrations might be stronger than the response after an initial adaptation time (Treydte et al., 2009; Drake et al., 1997). This may lead to lower $^{13}C$ discrimination and a decreasing correction factor. Moreover, anthropogenic warming increases the drought stress of plants growing in mid-latitude sites. To protect from drought stress, plants reduce stomatal conductance, which might lead to an increase in $\delta^{13}C_{LM}$ values (Büntgen et al., 2021). The modern correction of $\delta^{13}C_{LM}$ values, especially in the last decade, may overcorrect the raw $\delta^{13}C_{LM}$ values and lead to overrated modelled temperatures. It is thus important to note that there are many additional uncertainties beyond the linearity and the strength of the physiological response to increasing $CO_2$ concentrations.

The correlation coefficients between gridded instrumental and modelled summer temperatures are weak between 1935 to 1954 and 1939 to 1965 when using the $\delta^{13}C_{LM\_high\text{-}frequency}$ and $\delta^{13}C_{LM\_RL}$ values, respectively. During this early-to-mid 20th century period, the $\delta^{13}C_{LM}$ values seem to be influenced by more than one environmental factor, and correlation with a single climatic parameter like temperature is probably an oversimplification (McCarroll and Loader, 2004). Inconsistent controlling factors on

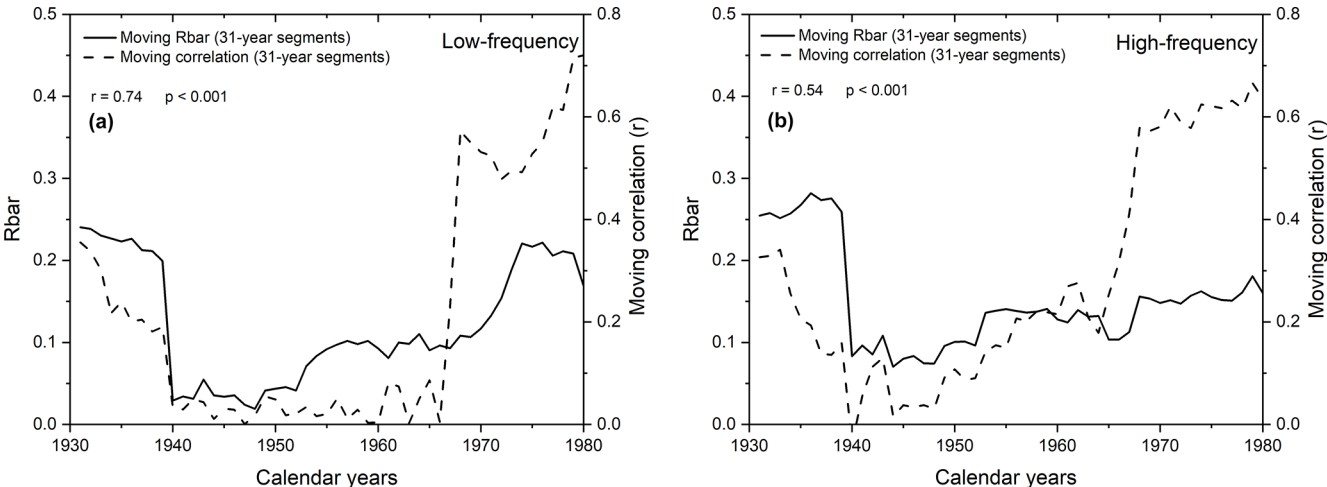

**Figure 10.** 31-year moving correlations between gridded instrumental and modelled summer temperatures using $\delta^{13}C_{\text{LM\_RL}}$ values **(a)** and $\delta^{13}C_{\text{LM\_high-frequency}}$ indices **(b)** (dashed lines) plotted together with the 31-year moving Rbar values of these data (solid lines).

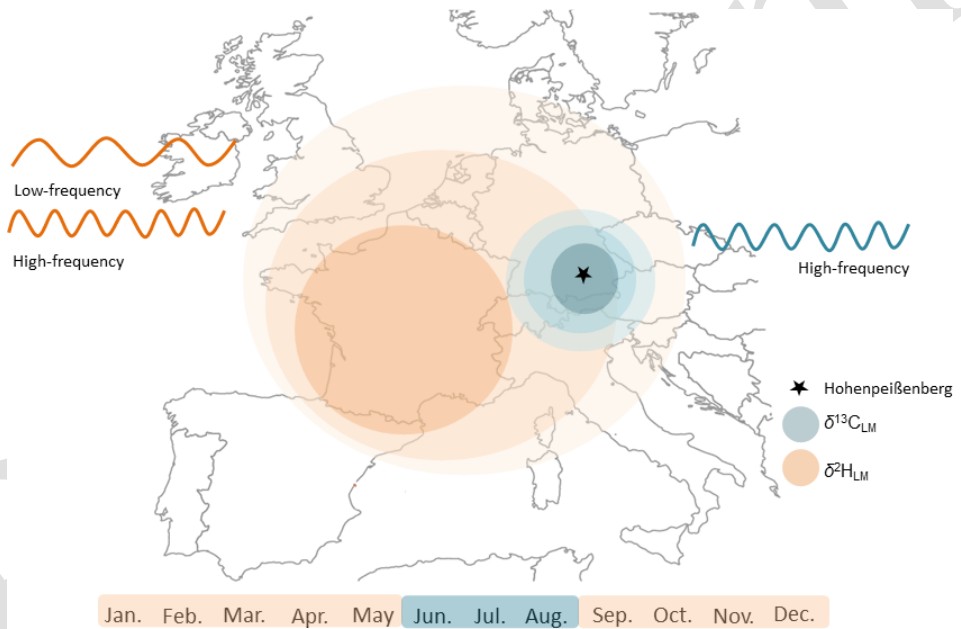

**Figure 11.** Schematic diagram illustrating the beneficial application of stable isotope values of lignin methoxy groups for temperature reconstructions. Stable carbon isotope values reflect mainly the regional summer temperatures and $\delta^{2}H_{\text{LM}}$ values the large-scale "shifted" annual temperatures. The combination of the two isotopic approaches supports temperature reconstructions on different temporal and spatial scales.

$\delta^{13}C_{\text{LM}}$ values would also be responsible for intra-series inconsistencies. This conclusion is supported by contemporaneous low Rbar values and low-moving correlation coefficients between the carbon isotope chronology and temperature (Fig. 10). Indeed, both chronologies correlate strongly ($r = 0.74$ for $\delta^{13}C_{\text{LM}}$ values and $r = 0.54$ for $\delta^{13}C_{\text{LM\_high-frequency}}$ indices).

As mentioned before (Sect. 4.1), there are several other factors apart from climate that may influence tree-ring $\delta^{13}C$ values.

In this study, however, we used two cores from just four trees each. Irregularities in only one or two trees massively affect our mean chronology. The study side at Hohenpeißenberg is located in an area that is strongly influenced by human activity – for example, soil sealing (a change in soil structure due to the covering of land, e.g.

forest roads) might result in reduced nutrients and water availability, whereas tree clearing could strongly increase irradiance. Such factors conceal the temperature signal (Saurer et al., 1997) and could be an explanation for the low Rbar values and the poor relationship between $\delta^{13}C_{LM}$ series and temperature prior to 1966.

Calculating the RE and CE statistics without the period of low inter-series correlation, the RE and CE values substantially increase, indicating the potential of temperature reconstructions by $\delta^{13}C_{LM}$ values if tree-ring series are not influenced by non-significant inter-series correlations and therefore mainly controlled by one environmental factor. The temporal robustness could be improved in further studies by using more replicates in sample sites with less human activity.

### 4.3   $\delta^2H_{LM}$ values and climate response

The recently provided correction method by Greule et al. (2021) improves the $\delta^2H_{LM}$ series of Anhäuser et al. (2020) as a regional temperature proxy. We found a correlation of $r = 0.58$ with local "shifted" annual temperatures (Fig. 9), which is slightly higher than the observation of Anhäuser et al. (2020). In addition, we confirm the strong correlation of $r = 0.69$ with western European surface temperatures as shown by Anhäuser et al. (2020). The increasing long-term trend of the $\delta^2H_{LM}$ series clearly reflects the anthropogenic warming trend (slope of $0.14\,\text{mUr}\,\text{yr}^{-1}$ from 1916–2015). However, a much stronger increase of $0.38\,\text{mUr}\,\text{yr}^{-1}$ was found during the most recent period from 1970–2015 (Fig. 8). Interestingly, the ratio between the two slopes of 2.7 is similar to the rates of gridded instrumental temperature changes of 2.4 over these two time periods ($0.015\,°\text{C}\,\text{yr}^{-1}$ from 1916 to 2015, and $0.036\,°\text{C}\,\text{yr}^{-1}$ from 1970 to 2015, Fig. S5). Hence, $\delta^2H_{LM}$ values can be used to position high- and low-frequency temperature changes within a long-term context (Fig. 11).

## 5   Conclusion

We measured the $\delta^{13}C_{LM}$ values of eight annually resolved 100-year *Fagus sylvatica* tree-ring series from the Hohenpeißenberg in southern Germany and evaluated their sensitivity to climate variations. $\delta^{13}C_{LM}$ values were corrected for the Suess effect and for the physiological response to increasing atmospheric $CO_2$ concentrations using different factors for possible changes in discrimination CE2. The highest correlations with temperature were recorded with $\delta^{13}C_{LM}$ values that were corrected for the Suess effect and with a correction factor that accounts for a strong $CO_2$ response of $0.032\,\text{mUr}\,\text{ppmv}^{-1}$ ($\delta^{13}C_{LM\_RL}$) as suggested by Riechelmann et al. (2016). At Hohenpeißenberg, inter-annual to decadal summer temperature variations are significantly reflected in $\delta^{13}C_{LM}$ values. The highest correlation was

observed between JJA temperatures and $\delta^{13}C_{LM\_RL}$ values at $r = 0.68$ ($p < 0.001$). Lower but still highly significant correlations were recorded for annual and "shifted" annual temperatures. To assess the temporal stability of our tree-ring proxy, summer temperatures were modelled by linearly regressing the $\delta^{13}C_{LM\_RL}$ chronology. Highly significant running correlations, particularly over the last 50 years, indicate the potential of $\delta^{13}C_{LM}$ values for reconstructing summer temperatures at annual resolution. The highly significant correlations between gridded instrumental temperatures and $\delta^{13}C_{LM\_high\text{-}frequency}$ values confirm the suitability of this proxy to reconstruct high-frequency summer temperatures. To reconstruct long-term trends with $\delta^{13}C_{LM}$ values, a further understanding of how plant isotopic discrimination changes due to increasing $CO_2$ concentration is essential.

However, our results also indicate that temperature reconstructions based on stable isotope ratios of tree-ring lignin methoxy groups are sensitive to low inter-series correlations. These uncertainties were quantified by evaluating moving Rbar values, RE, and CE statistics and can be improved in further studies by increasing the number of replicate tree samples.

Additional consideration of $\delta^2H_{LM}$ values from the same trees (Anhäuser et al. 2020; corrected after the suggestion of Greule et al., 2021) demonstrate that $\delta^2H_{LM}$ values predominantly reflect large-scale temperatures, since the highest correlations were found with western European "shifted" temperatures ($r = 0.69$, $p < 0.001$) and somewhat lower correlations with local "shifted" temperature variations ($r = 0.58$, $p < 0.001$).

The results obtained in this study described, for the first time, a reliable summer temperature proxy derived from $\delta^{13}C_{LM}$ values in temperate, low elevation environments. We found that $\delta^{13}C_{LM}$ values perform better with regional and $\delta^2H_{LM}$ values with large-scale temperatures, indicating that the two proxies could be combined to reconstruct long-term temperature variations at different spatial and temporal scales.

**Data availability.** We provide the data in heiDATA, which is an institutional repository for research data of the University Heidelberg (https://doi.org/10.11588/data/ZCMVUY TS2, Wieland et al., 2022).

**Supplement.** The supplement related to this article is available online at: https://doi.org/10.5194/cp-18-1-2022-supplement.

**Author contributions.** FK, AW conceived the study. MG performed the measurements and analysed the data together with FK and AW. JE and PR assisted the application of detrending methods and helped to place the results in a wider dendroclimatological context. The paper was written by FK, AW, JE, PR, and MG. All authors also have given approval to the final version of the manuscript.

**Competing interests.** The contact author has declared that none of the authors has any competing interests.

**Disclaimer.** Publisher's note: Copernicus Publications remains neutral with regard to jurisdictional claims in published maps and institutional affiliations.

**Acknowledgements.** We are grateful to Tobias Anhäuser, Birgit Sehls, Bernd Knape, Claudia Hartl, and Werner Thomas for support in collecting and preparing wood samples.

**Financial support.** This research has been supported by the German Research Foundation DFG (KE 884/6-3, KE 884/8-2, KE 884/17-1). Philipp Roemer received support from the German Research Foundation (grant no. ES 161/12-1) and Jan Esper from ERC Advanced Grant Monostar (AdG 882727) and SustES: Adaptation strategies for sustainable ecosystem services and food security under adverse environmental conditions (grant no. CZ.02.1.01/0.0/0.0/16_019/0000797).

**Review statement.** This paper was edited by Hans Linderholm and reviewed by two anonymous referees.

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

## Remarks from the language copy-editor

## Remarks from the typesetter