# Peer review of "Climate signals in stable carbon and hydrogen isotopes of lignin methoxy groups from southern German beech trees"

_Climate of the Past, 2021_

## Author Response (AR1)

**Point-by-point response to the issues raised by Referee #1**

We thank Referee #1 for the positive evaluation of our work and for the helpful comments to improve the manuscript. All comments and requested changes were taken into account. Please note that comments by the referee are in italics and that in the authors' answer the mentioned line numbers refer to the version of the revised manuscript including track changes.

*Wieland et al. present an interesting new annually resolved series of lignin methoxy d13C tree ring series. These are highly novel methodologies and promising for paleoclimate reconstructions.*
*While the records themselves are interesting – and should be published - I do have several concerns regarding the methods used to correct for the plant physiological effects and the interpretation of the long-term trends. I will first describe my main concern and then point out several smaller comments on the manuscript.*

**Main concerns:**

*d13C in plant material is strongly influenced by various environmental and ecophysiological factors. These include*
*i) changes in atmospheric d13C and atmospheric CO2 concentration*
*ii) changes in atmospheric deposition in nitrogen*
*iii) change in tree light environment*
*iv) change in tree height*
*I will first focus on the effect of atmospheric d13C and atmospheric CO2 concentration changes. The authors correct for the d13C atmosphere effect or the Suess effect. That is all fine. However, the authors then move on to correct for the effect of plant physiological responses to atmospheric CO2 (eg. change in discrimination or iWUE) using several correction factors that have been proposed by various authors (Kurschner, Feng, Treydte) and which differ almost 3 fold in magnitude. The authors also use a correction factor developed by a previous study for higher altitude Larch trees (Riechelman et al. 2016). As shown in fig. 3 these corrections result in very different upward curves since ca. 1950 with some showing very strong increases in the "corrected" d13C.*
*I do not disagree with the need to correct for the effect of CO2 on these series, but we do not know enough about tree responses to CO2 to know which one of these "corrections" represents the "real" tree response. None of the corrections in the literature seem to argue in a particular convincing way how trees respond to CO2 and some just fit curves that results in the highest correlations with the targeted climate variable. In addition, tree ring d13C studies show that trees respond differently between sites and species.*
*In short, I cannot see how one can choose from these relative arbitrary correction curves which one is the best. The authors are favouring the correction from Riechelman as that results in the highest correlation with observed temperature (fig. 4, 5 and 6), but this is somewhat circular in my opinion. You add several artificial increasing trends to the d13C and then relate it to a climatic record of which we know that is has a positive trend and find a good match. But what do we really learn from this, and secondly can you use such a record for reliable climate reconstructions?*
*Several of the conclusions are entirely due to this methodological choice of adding trends to the d13C curve. For example, the increase in strength of the correlation with temperature for the upward corrected curve (fig. 4,5) is simply due to the addition of a trend to the series. It is also not surprising that the series with the strongest trends added, results in the strongest inter-series correlation (lines 207 etc). And again the d13C corrected according to Riechelmann, results in a good correlation with d2H as you have two series with strong upward trends (fig. 10), but the correlations vary in reality between negative (with the raw data) to slightly positive*

*when correcting for the Suess effect. In my view, we are not learning much from this, and I do not believe one can use these records put recent temperature increases in a longer term context. It seems to violate the stationarity principle and the correction for that is artificial. But do please correct me if i see this wrong.*

*One needs to know in much greater detail how CO2 truly affects plant isotope discrimination. In perspective of this and the poor correlations pre-1965, I wonder if the conclusion that "this is a suitable proxy for reconstructing high to low frequency summer temperatures" (lines 317). This is perhaps true for the high-frequency variation since 1966, but not for the low-frequency variation and not for the full period.*

> Authors: Following the referee's suggestion, the relationship between increasing correlation coefficients and applied correction methods as well as their limitations are now discussed in greater detail in section 4.1.
>
> Furthermore, the increasing Rbar values and correlation coefficients between $\delta^{13}C_{LM}$ and temperature are now described as a result of the different trends added to the $\delta^{13}C_{LM}$ series. In the revised manuscript, the correction factor of Riechelmann et al. (2016) is still considered but not indicated as the 'real' or 'ideal' correction factor.
>
> In addition, the summarizing sentence in section 5 '…the suitability of this proxy to reconstruct high-to-low frequency summer temperatures' has been modified to read '…the suitability of this proxy to reconstruct high-frequency summer temperatures. To reconstruct long-term trends with $\delta^{13}C_{LM}$ values, a further understanding of how plant isotopic discrimination changes due to increasing $CO_2$ concentration is essential.' (Line 518-520)
>
> Finally, section 3.5 'Comparison of $\delta^{13}C_{LM}$ and $\delta^2H_{LM}$ chronologies' was deleted from the manuscript. As the referee mentioned, the correlation coefficients between the two isotopic series are mainly a result of the applied correction procedures. This study showed, that $\delta^2H_{LM}$ values are mainly affected by large-scale MAT changes, whereas $\delta^{13}C_{LM}$ values are predominantly controlled by local summer temperature. Therefore, both isotopic ratios are not predominantly controlled by the same climate parameter, and we can thus not assume a strong correlation between the two series.

*My other main concern is that other factors that affect d13C are poorly discussed. This includes above mentioned effects of eg. Nitrogen deposition (see Leonardi et al. 2012), and effects of tree height and light (Brienen et al 2017, Vadeboncoeur et al. 2020). For these beech trees these may be very important factors that control tree isotope discrimination, but it depends on the size and age of trees. Such information needs to be added to this article and discussed. In fact, changes of climate responses with tree height could also well explain the poor relationship between d13C and temperature before 1966. For example, Trouiller et al. 2019 find that large and small tree differ in their growth response and one could thus also expect that the response of d13C will differ.*

> Authors: Good point! Further information on how additional factors could affect $\delta^{13}C_{LM}$ values were added to the manuscript in section 4.1 from line 393. In addition, the effects of an age-related trend, including tree size and age, were excluded by calculating a site-specific linear regression model between trees with different ages over a common period (see section 4.1 from line 400).

**Minor comments:**

*In the introduction in lines 51-61 ... Can you expand the section on d2H a bit more and say where the signal comes from (source water, leaf enrichment or both), if this is known.*

Authors: Recently, Greule et al. (2021) provided detailed information about the biosynthetic pathway responsible for $\delta^2$H fractionation between precipitation and lignin methoxy groups in tree rings. In this study, it is considered that precipitation accumulates to soil water and further to xylem water with no substantial isotope fractionation. Isotopic fractionation occurs in mainly two biosynthetic processes, the transfer reaction of serine to the $CH_2$-unit (ranging from 0 to -50 mUr) and the hydrogen atom transfer by certain flavoproteins with a depletion of -580 mUr down to -780 mUr (for further information, please refer to the study by Greule et al. (2021)). Based on the mentioned constraints, $\delta^2$H$_{LM}$ values are directly connected to the source water and likely not influenced by further isotope fractionation such as leaf water enrichments. Further information about the origin of the $\delta^2$H$_{LM}$ signal has been added to the revised manuscript from line 72.

*Section 71-79: Some of the statements are a bit over assertive: Do we really know that much about mesophyll conductance and the effects of Ca on photorespiration to make these statements? Be more careful here as there are large uncertainties with the variables in eq. 1.*

Authors: We added further information in the revised manuscript (line 99 and 100).

*Line 82: better to say .. "stomatal control limits photosynthesis" (cannot say gs is higher than the rates of photosynthesis),*

Authors: Change applied (line 106).

*Lines 80-86: perhaps also mention post-fractionation processes?*

Authors: Further information about post-photosynthetic fractionation was added (line 109-114).

*Fig. 1: add proper units to the precipitation axis that can be understood… eg. mm per day or mm per month.*

Authors: Change applied.

*Lines 183: Bravais Person?? Pearson correlation coefficient*

Authors: Change applied (line 213).

*Lines 211: What is the low frequency series? The LM_R as in figure 5? Why is the LM_RL a low frequency series?*

Authors: The low-frequency series is referred to the $\delta^{13}$C$_{LM}$ values. However, the referee is correct that this declination might be misleading in this context. For clarification, the term 'the raw $\delta^{13}$C$_{LM}$' was added to the manuscript (line 245) and the subtitle of Figure 2 was adjusted.

*Equations 3 and 4 are not clear. They are the same but for different periods or is this for different series? Please explain.*

> Authors: Equations 3 and 4 contain the same period but summer temperatures were modelled by two different series the $\delta^{13}C_{LM\_RL}$ following Equation 3 and the $\delta^{13}C_{LM\_high-frequency}$ series following Equation 4. The equation results from the linear regression between the isotope ratios ($\delta^{13}C_{LM\_RL}$ or $\delta^{13}C_{LM\_high-frequency}$) and the summer temperatures during the period 1916-2015.

*Line 341: Higher compared to what? To other species? Are you talking about higher mean discrimination, or higher changes in discrimination over time (i.e. a steeper increase in discrimination)?*

> Authors: Compared to other species. The correction factors of Feng and Epstein (1995) and Treydte et al. (2009) were determined to mostly evergreen conifers. Riechelmann et al. (2016) used larches, which are deciduous coniferous trees. Secondly, we mentioned a higher mean discrimination. The correction factor of 0.032 mUr year$^{-1}$ is constant and therefore independent of increasing $CO_2$ concentrations.
> We have made this clearer in the revised manuscript (from line 389).

*Line 343: strange statement … "It has been shown .. "*

> Authors: Sentence has been revised (line 391).

*Line 351: why is this not due to a decrease in gs due to increase in RH or VPD with increasing T? Are we also seeing a positive T response in the tree ring widths? Please discuss this.*

> Authors: The referee is correct, the relative humidity (RH) correlates with the temperature and the vapour pressure deficit (VPD) and directly controls stomatal conductance (gs).
> The temperature response in $\delta^{13}C_{LM}$ values could be at least part of several indirect signals since factors like RH, soil moisture status, and drought stress also strongly correlate with temperature. However, RH, soil moisture status, or drought stress are also strongly correlated to antecedent precipitation, and we cannot see any precipitation signal in our $\delta^{13}C_{LM}$ series. Thus, we interpreted the strong temperature and weak precipitation signals as an indication that $\delta^{13}C_{LM}$ values are predominantly controlled by the photosynthetic rate.
> Further information was added in the revised manuscript (line 424-432).
> We observed only weak correlations between temperatures and TRW data (for example correlations between high-frequency series: $r_{summer} = -0.08$; $r_{MAT} = 0.14$; $r_{prev.Sep.-Aug.} = 0.10$). This might be explained because the study site (Hohenpeißenberg) is situated at mid-latitude and rather low elevation environment. Typically, good correlations between TRW and temperature have been reported for trees at growing boundaries, at high latitude or far north.

*Lines 357-363: Explain this a bit more. Trees were supposedly younger, smaller in pre 1966, Could that explain the change? Trees were perhaps below the canopy and limited by other factors? Please discuss further.*

> Authors: Age-related trends, including tree age and hight, could not be reported in this study (Further explanation see section 4.1 from line 400 in the revised manuscript).

Further explanation of how different environmental factors might have influenced the $\delta^{13}C_{LM}$ series before 1966 were added to the revised manuscript from line 459.

*Lines 368-369: drought stress i only mentioned here for the first time. Why? include this possible mechanism also in earlier sections. It is not just Assimilation that affects d13C. And you might be able to check if d13C is controlled by A or gs when you also include analysis of ring width. If ring width increases in line with d13C then it must be assimilation controlled, if it is the opposite then it must be controlled by gs.*

Authors: Drought stress is now mentioned as one environmental factor that influences carbon isotope fractionation. This part has been moved to section 4.1. Moreover, comparison between tree ring width and $\delta^{13}C_{LM}$ values support our assumption that $\delta^{13}C_{LM}$ values are dominantly controlled by the assimilation rate. We added further information on this in section 4.2 from line 424 and included a figure showing the relationship between TRW and $\delta^{13}C_{LM}$ series in the supplemental (S4).

*Line 370: you mean overcorrect the original (raw d13C)?*

Authors: Correct. For clarification purposes 'raw $\delta^{13}C_{LM}$ values' was added (line 452).

*Line 371: indeed a lot of uncertainties that can move your recent trends in d13c any direction depending on the uncertainties.*

Authors: Yes, we agree. The multitude of uncertainties potentially affecting $\delta^{13}C_{LM}$ values have now been adequately highlighted in the revised manuscript (from line 393 and from line 465).

*Line 375: inter-series inconsistencies in the early part of the record again indicate that other factors than climate affect d13C.*

Authors: Further explanations were added to section 4.2 from line 459.

*Line 379-380 "additionally .. " explain a bit more. What is soil sealing?*

Authors: Additional factors were added, and soil sealing explained in more detail (line 468).

*Line 397 .. intensified anthropogenic warming .. this is not clear. What do you mean why do you say intensified? Is that in comparison with the temp increase? The trends in temp and in d2H look pretty similar to me, and no need perhaps for other factors to be involved than simply temperature.*

Authors: To avoid any confusion but also not to overinterpret the impact on additional environmental factors we have removed the section dealing with drought from the manuscript.

**Point-by-point response to the issues raised by Referee #2**

We thank Referee #2 for the positive evaluation and the helpful comments. All comments and requested changes were taken into account. Please note that comments by the referee are in italics and that in the authors' answer the mentioned line numbers refer to the version of the revised manuscript including track changes.

**General comments**

*As reviewer 1 states, tree-ring lignin methoxy d13C and d2H is an interesting and new methodology, and it may offer additional information about past climate compared to traditional tree-ring widths, density and cellulose isotope ratios, and therefore readers of this journal may find this manuscript worthwhile. However, discussion of this manuscript misses some important points readers may wish to know. As a non-specialist of tree-ring lignin methoxy groups, I wanted to read more about the comparison between traditional tree-ring cellulose isotope ratios and tree-ring isotopes of lignin methoxy groups, however, such comparison was not sufficiently described in the manuscript.*

> Authors: We have added further information regarding isotopic signatures of cellulose and lignin methoxy groups in trees. For more details, please refer to the answers to the specific comments below. Please note that the development and application of lignin methoxy groups as a paleoclimate proxy is a relatively young research field and so far, there exist only a limited number of studies, which have measured and compared isotopic values of methoxy groups and cellulose of the same tree rings.

*I think majority of the readers of this manuscript will be those who study tree-ring isotopes and therefore I think authors should add more information about the difference between the two. For example, compared to cellulose deposition, lignin deposition to cell walls happens at the latter part of the growing season and duration of lignin deposition to cell walls is longer than that for cellulose, which may explain the correlations observed in this study (Ln 16-19). Authors should discuss more about the physiological background of the correlations observed in this study.*

> Authors: As mentioned in our response before, we have added more information to the introduction. To deal with this issue in much greater detail in the discussion section, it would have been necessary to compare the isotopic ratios from lignin methoxy groups with those derived from cellulose of the same trees at Hohenpeißenberg. Due to labour, time, and financial aspects, we were not able to also measure the isotopic ratios of cellulose from these trees. Thus, it makes not much sense to compare the results of lignin methoxy groups from Hohenpeißenberg with previous studies, where also cellulose has been measured. However, for a detailed discussion regarding the comparison of stable isotope values of methoxy groups with cellulose or bulk measurements we refer the referee and the readers to the studies by Mischel et al. (2015) and Gori et al. (2013). Concerning the specific question (Ln 16.-19), please refer to the answer below (specific comments). We have added further information regarding the observed differences between the carbon isotope composition of cellulose, whole wood, and lignin methoxy groups to the revised manuscript (from line 51).
>
> In addition, we included a new schematic diagram to the revised manuscript (Figure 11) to illustrate to readers the novel application of $\delta^{13}C$ and $\delta^2H$ values of lignin methoxy groups for temperature reconstructions. Stable carbon isotope values reflect mainly the regional summer temperatures, whilst $\delta^2H_{LM}$ values show large-scale 'shifted' annual temperatures. The combination of the two isotopic approaches provides information on different temporal and spatial scales. To the best of our knowledge, this is not possible with 'traditional' stable isotope tree ring analyses such as from cellulose.

**Specific comments**

*Ln 16-19 "The calibration of δ13CLM chronologies against instrumental data reveals highest correlations with regional summer (r = 0.68) and mean annual temperatures (r = 0.66), as well as previous-year September to current-year August temperatures (r = 0.61)"*
*What about EPS? Did you or any previous studies compared traditional tree-ring isotope ratios and tree-ring isotopes of lignin methoxy groups?*

Authors: For the 1916-2015 period, the calculated EPS value of the $\delta^{13}C_{LM}$ chronology is 0.71. The study by Anhäuser et al. (2020) analysed the $\delta^2H_{LM}$ values of the same tree ring sample set and recorded an EPS value of 0.91, indicating a sufficient sample size to establish a representative $\delta^2H_{LM}$ chronology at Hohenpeißenberg. It should be noted that the study by Anhäuser et al. (2020) included 9 different cores. F1 was represented by three cores and F2-F4 by only two cores. To eliminate a higher quantifier of F1, all $\delta^{13}C_{LM}$ chronologies of the four trees are now represented by two cores.

In several studies the threshold of $\geq 0.85$ for the EPS value is controversial discussed (for instance: Wigley et al., 1984; Briffa and Jones 1990; Mérian et al., 2013). Buras (2017) concluded that the application of the EPS does not necessarily give us a valid information whether tree ring data can be used for climate reconstructions. However, in this study we used eight cores from four different trees and as we described in the manuscript (line 480 and 523), that ideally more replicates would be required for $\delta^{13}C_{LM}$ studies to produce a temporally robust mean $\delta^{13}C_{LM}$ chronology. In this case, a larger number of replicates would also lead to an increased EPS value.

The EPS value has been added and discussed in the revised manuscript from line 227.

The study by Mischel et al. (2015) compared the oxygen, hydrogen, and carbon isotopic composition of whole wood, alpha-cellulose, and lignin methoxy groups. In this study, EPS analysis showed that four trees were sufficient for $\delta^{13}C$ measurements of whole wood, and three to four trees were adequate for $\delta^{13}C$ measurements of cellulose. For $\delta^{13}C_{LM}$ measurements, the sample set should be increased to seven replicates. Nevertheless, with this study we could show that $\delta^{13}C_{LM}$ values have a great potential for reconstructing temperature changes if tree ring series are not influenced by inconsistencies within the series.

In addition, the study by Gori et al. (2013) analysed a similar approach and compared hydrogen, oxygen, and carbon isotope compounds of whole wood, cellulose, and lignin methoxy groups. This study used tree samples from three different elevation sites in the south-eastern Alps (900 m, 1300 m, and 1900 m) and showed that the carbon, oxygen, and hydrogen isotopic ratios of whole wood and cellulose were highly correlated while the carbon and hydrogen isotopic composition of lignin methoxy groups correlate to a lesser extent with the other components. Similar findings were documented in the study by Mischel et al. (2015). They analysed trees from a low elevation environment (Altenkirchen, Germany, about 300 m). Here, the $\delta^{18}O$ and $\delta^{13}C$ values of whole wood and cellulose were also highly correlated (r = 0.89; r= 0.96) while the correlations between the carbon isotope values of whole wood and lignin methoxy groups or cellulose and lignin methoxy groups showed the same but lower correlation coefficient of r = 0.72. The two studies suggest that stable carbon and hydrogen isotope values of methoxy groups contain a different climate signal and seem to be influenced by different environmental and biochemical factors. The extraction of cellulose, however, may not be necessary as the isotopic compounds of cellulose and whole wood receive the same climate signal. The isotope measurement of lignin methoxy groups, on the other hand, might be a beneficial proxy for climate reconstructions in a different temporal and spatial context.

This conclusion is supported by the study of McCaroll et al. (2003) which implicated that the key to amplifying the climate signal lies in combining independent proxies that are not similar. In this context, Gori et al. (2013) showed that the best prediction model for reconstructing temperature changes is obtained when the hydrogen and carbon isotope compounds of whole wood and methoxy groups are combined.

We have added further information regarding the observed differences between carbon and hydrogen isotopic composition of cellulose, whole wood, and lignin methoxy groups to the revised manuscript (from line 51). However, for detailed discussion regarding comparison of stable isotope values of methoxy groups with cellulose or bulk measurements, we refer the referee and the readers to the studies by Mischel et al. (2015) and Gori et al. (2013).

*Ln 24 "large-scale temperatures" Why dHLM reflects larger scale temperature than dCLM? Can this be explained partly by the fact that hydrogen isotope ratios of precipitation reflects larger scale temperature of this area?*

Authors: The $\delta^2H_{LM}$ values are directly connected to the $\delta^2H$ values of the precipitation. For detailed information about the biosynthetic pathway which is responsible for $\delta^2H$ fractionation between precipitation and lignin methoxy groups in tree rings please refer to the study by Greule et al. (2021). Therefore, the referee is right, that the stable water isotopes of precipitation of this area are considered to indicate large-scale atmospheric phenomena rather than variations in local or regional climate states.

Further information has been added to the revised manuscript from line 82.

*Ln 31-32 "Weather and climate parameters affect the physiological process within these tree rings."*
*As far as hydrogen isotope ratios are concerned, weather and climate (temperature) also affects hydrogen isotope ratios of precipitation, which eventually affects tree ring d2H. This is not physiological, but a hydrological process.*

Authors: The sentence has been revised to read: 'Weather and climate parameters affect direct or indirect the physiological process within these tree rings.' (Line 31)

*Ln 54-55 "Therefore, the temperature dominated signal in δ2Hprecipitation (Dansgaard, 1964) is reflected in δ2HLM values as has been demonstrated for mid-latitude sites (Anhäuser et al., 2017a; Greule et al., 2021)."Is d2H precipitation not affected by the amount of precipitation (of a single precipitation event)? Please explain.*

Authors: Yes, the isotopic composition of precipitation is also influenced by the amount of precipitation. The amount effect describes lower $\delta^2H_{precip}$ values in rainy months and higher $\delta^2H_{precip}$ values in months of less rainfall. This effect occurs mainly in tropical regions as well as in high alpine mountain ranges. However, in mid- and high latitudes, the temperature effect is the dominating factor controlling the isotopic composition of precipitation (latitude effect). In this context, several studies suggested that $\delta^2H_{LM}$ values reflect $\delta^2H$ values of the tree's source water as significant relationships between $\delta^2H_{LM}$ values and mean annual $\delta^2H_{precip}$ values have been observed implying that $\delta^2H$ values of the tree's source water reflect an annual integral of site-specific $\delta^2H_{precip}$ values. We added more information to the revised manuscript (line 76-78).

*Ln 57-60 "Wang et al. (2020) found significant correlations between δ2HLM values and April-August temperatures (r = 0.58 to 0.7) for two coniferous species (Larix gmelinii, larch and Pinus sylvestris var. mongolica, pine) from a permafrost forest in northeastern China."*
*Lignin deposition takes place over a longer period of time than cellulose deposition. Therefore, I expect tree-ring lignin to reflect temperatures over a longer period of months compared to tree-ring cellulose from the same tree. April-August seems longer than those for tree rings,*

*which usually shows correlations to summer (June-July) climatic variables in my impression. If such comparison is possible, then please add more explanation here about the difference between cellulose and lignin.*

> Authors: The studies of Mischel et al. (2015) and Gori et al. (2013) suggested that the isotopic composition of methoxy groups reflects different environmental and biochemical factors than the isotopic composition of cellulose or whole wood. As the referee mentioned one possible explanation could be that the lignin deposition occurs over a longer time period than cellulose deposition. In contrast to this assumption, Mischel et al. (2015) found the highest correlation between $\delta^2$H values of lignin methoxy groups and maximum temperatures at the beginning of the growing season, while $\delta^{18}$O values of cellulose from the same sample set showed the highest correlation with the maximum temperatures from March to October. Gori et al. (2013) found the highest correlation between $\delta^2$H values of cellulose and seasonal and mean annual temperatures where these correlations are strongly dependent on the altitude of the sample sites. This study applies a temperature proxy from a low elevation environment. The study by Wang et al. (2020) analysed trees from a permafrost forest and most of the 'traditional' isotope ratio analysing studies used trees from growing boundaries, at high latitude or on high elevation sites. Sample site specific factors like snowmelt, monsoon events, growing period, or different growing limitation factors massively influence the temperature sensitivity. It is, therefore, difficult to compare the temperature sensitivity between the proxies from different studies.
>
> In addition, if tree ring lignin reflects temperatures over a longer period of months compared to tree ring cellulose, the $\delta^{13}$C$_{LM}$ analysis in this study should also reflect temperatures over a longer period. Indeed, our results showed that the $\delta^{13}$C$_{LM}$ values are highly correlated with summer temperatures (June, July, and August) and thus similar to the results of the cellulose-based analysis.
>
> Consequently, we assume that varying site-specific factors make it difficult to compare the temperature sensitivity of cellulose, whole wood, or lignin methoxy groups. It will be necessary to measure the isotope ratios of cellulose and lignin methoxy groups from the same sample set in further studies to understand the impact of cellulose and lignin deposition.

*Lines 63-64 "The carbon of each annual tree ring has its origin in the atmospheric CO2" What about the origins of hydrogen of each annual tree ring? Please explain.*

> Authors: Recently, Greule et al. (2021) provided detailed information about the biosynthetic pathway responsible for $\delta^2$H fractionation between precipitation and lignin methoxy groups in tree rings. In this study, it is considered that precipitation accumulates to soil water and further to xylem water with no substantial isotope fractionation. This is supported by a recent study of Chen et al. (PNAS, 2020). Isotopic fractionation occurs in mainly two biosynthetic processes, the transfer reaction of serine to the $CH_2$-unit (ranging from 0 to -50 mUr) and the hydrogen atom transfer by certain flavoproteins with a depletion of -580 mUr down to -780 mUr (for further information, please refer to the study by Greule et al. (2021)). Based on the mentioned constraints, $\delta^2$H$_{LM}$ values are directly linked to the source water and likely not influenced by further isotope fractionation such as leaf water enrichments.
>
> See also the response to referee #1, regarding a similar comment.

*Ln 137-138 "The maximum differences between two individual cores of the same tree ranged from 1.54 for F1 to 3.26 mUr for F2"Did you calculate expressed population signal for*

*d13CLM? If so, how was it compared to tree rings from the same trees? Is d13CLM more coherent than d13C of tree rings?*

> Authors: The mean EPS value of the $\delta^{13}C_{LM}$ chronology was 0.71. Correlations between the $\delta^{13}C_{LM}$ values of two cores of one tree were 0.73 for F1 (EPS: 0.85), 0.54 for F2 (EPS: 0.701), 0.6 for F3 (EPS: .075), and 0.31 for F4 (EPS: 0.47). The EPS signal of the mean $\delta^{13}C_{LM}$ chronology is thus similar to the two series of one tree.

*Ln 418-419 "However, our results also indicate that temperature reconstructions based on stable isotope ratios of tree ring lignin methoxy groups are sensitive to low inter-series correlations." Does this mean inter-series correlations for lignin methoxy groups were lower than those for tree-ring cellulose? Please write more about the comparison between carbon and hydrogen isotope ratios of tree-ring lignin methoxy groups and tree-ring cellulose, because it is an important information for readers to judge if analysis of tree-ring lignin methoxy groups are worth trying.*

> Authors: We assume that sample site-specific factors are responsible for our low inter series correlations between 1940 to 1965. This study used two cores from only four trees. Irregularities in only one or two trees would massively influence our inter series signal. The study site Hohenpeißenberg is massively influenced by human activities. Factors such as tree clearing, nitrate, or water availability may conceal the climate influence factor (further limiting factors were discussed in section 4.2 of the manuscript). As these factors are strongly dependent on the sampling location it is difficult to compare the Rbar values of this study with other studies. The referee is certainly right, it would be interesting to compare the inter series correlation of the isotopic ratios from cellulose and lignin methoxy groups from the same sample set. Due to labour, time and financial aspects, we were not able to also measure the isotopic ratios of cellulose at this site so far.
> However, there are several benefits regarding stable isotope measurements of methoxy groups if compared with other isotope proxies such as cellulose. These are:
> - Bulk wood samples can be used
> - No time-consuming extraction procedures of single wood structural compounds
> - Removal of water from the samples is not necessary
> - For hydrogen isotope measurements no hydrogen exchange of covalent bound methoxy hydrogen atoms with surrounding water has been reported
> For detailed discussions of these issues, we would like to refer the referee to the studies by Anhäuser et al. (2020), Greule et al. (2021, 2009, 2008), and Keppler et al. (2007).
>
> For a detailed comparison of isotope values of cellulose and methoxy groups, we would like to refer the referee again to the studies by Mischel et al. (2015) and Gori et al. (2013). These researchers show that isotopic ratios of cellulose and whole wood are highly correlated, whilst the isotopic ratios of lignin methoxy groups seem to be also controlled by different environmental and physiological factors. Measuring $\delta^{13}C_{LM}$ and $\delta^2H_{LM}$ values therefore provide us a more time-saving method to analyse additional past climate changes in a different spatial and temporal context.
>
> Further information was added to the revised manuscript from line 51.

*Figure 2 Why does moving Rbar suddenly decreases around 1940? Is it possible that one of your samples are wrongly dated after 1940?*

> Authors: We assumed that site specific factors influence the robustness of the mean inter series correlation (See chapter 4.2). The TRW measurements show no indication that one of the samples were wrongly dated after 1940.

*Figure 5 If you calculate temperature from Hohenpeissenberg with temperature all over Europe, then I think you will find similar results, because temperature shows similarity over wider regions compared to precipitation. So I think this is self-evident. What do you think?*

Authors: Interesting point! The local temperature at Hohenpeißenberg is highly correlated to the temperature changes mostly all over Europe. So, $\delta^{13}C_{LM}$ values also show a good correlation with large-scale temperature changes. However, having a closer look at Figure 5 shows us that $\delta^{13}C_{LM}$ values are most strongly correlated with local temperature variations. If we now compare this with the $\delta^2H_{LM}$ chronology, we can see that the $\delta^2H_{LM}$ values are mainly controlled by large-scale atmospheric phenomena. Anhäuser et al. (2020) found that atmospheric circulation such as the NAO index, meridional atmospheric transport, or the potential mixing of air masses influence the isotope signal in precipitation and thus directly the $\delta^2H_{LM}$ values. Therefore, we would like to keep Figure 5 in the revised manuscript.

**Technical corrections**

*Ln 13 "skilful" sounds strange. What about "less successful"?*

Authors: Change applied (line 13).

*Ln 37 "For a more detailed overview of its applications in paleoclimate research, readers can refer to following studies"*

Authors: Change applied (line 37 and 38).

*Ln 41 Change " a-cellulose" to "α-cellulose" or "alpha-cellulose".*

Authors: Change applied (line 42).

**References**

Greule, M., Wieland, A., Keppler, F., 2021. Measurements and applications of δ2H values of wood lignin methoxy groups for paleoclimatic studies. Quat. Sci. Rev. 268, 107107.

Chen, Y., Helliker, B.R., Tang, X., Li, F., Zhou, Y. and Song, X., 2020. Stem water cryogenic extraction biases estimation in deuterium isotope composition of plant source water. Proceedings of the National Academy of Sciences 117, 33345-33350.

Gori, Y., Wehrens, R., Greule, M., Keppler, F., Ziller, L., La Porta, N., Camin, F., 2013. Carbon, hydrogen and oxygen stable isotope ratios of whole wood, cellulose and lignin methoxyl groups of Picea abies as climate proxies. Rapid Commun. Mass Spectrom. 27, 265–275.

Mischel, M., Esper, J., Keppler, F., Greule, M., Werner, W., 2015. δ2H, δ13C and δ18O from whole wood, α-cellulose and lignin methoxyl groups in Pinus sylvestris: a multi-parameter approach. Isotopes Environ. Health Stud. 51, 553–568.

McCarroll, D., Jalkanen, R., Hicks, S., Tuovinen, M., Gagen, M., Pawellek, F., Eckstein, D., Schmitt, U., Autio, J., Heikkinen, O., 2003. Multiproxy dendroclimatology: A pilot study in northern Finland. Holocene 13, 829–838.

---

## Author Response (AR2)

**Point-by-point response to the issues raised by Referee #1**

We would like to thank Referee#1 for revaluating our manuscript and proposing it for final publication.

*The authors did an excellent job at responding to my review and it is now a nice well balanced manuscript that reflects well the various uncertainties regarding the interpretation of the trends and the CO2 response of trees.*
*I suggest to accept this ms with a few last, very minor corrections.*

**Corrections:**
*Line 388 (of version with tracked changes): reword this sentence. The wording "suggests" make it sound as if this emerges for the data while it simply emerges from the artificial trends imposed. Thus say instead: "The use of this correction factor is amongst the strongest in literature for trees growing in low elevation environments and would constitute a strong CO2 response... "*
    Authors: Sentence has been revised (line 364 and 365).

*line 397: change annual nitrate rates to "nitrogen deposition"*
    Authors: Change applied (line 373).

*line 106: the RATE OF photosynthesis*
    Authors: Change applied (line 105).

*Specify still the equation 4. It is still unclear in the ms what eq 4 refers to.*
    Authors: To clarify the content, we have added further description in the equation (line 288, Eq. 4).

*line 529: preform = performs?*
    Authors: Change applied (line 483).

*line 426 remove "are"*
    Authors: Change applied (line 400).